# Universal Value-Function Uncertainties

**Moritz A. Zanger, Max Weltevrede, Yaniv Oren, Pascal R. Van der Vaart,**
**Caroline Horsch, Wendelin Böhmer & Matthijs T. J. Spaan**
Department of Intelligent Systems, Delft University of Technology
Delft, 2628 XE, The Netherlands
Correspondence to: `m.a.zanger@tudelft.nl`

## Abstract

Estimating epistemic uncertainty in value functions is a crucial challenge for many aspects of reinforcement learning (RL), including efficient exploration, safe decision-making, and offline RL. While deep ensembles provide a robust method for quantifying value uncertainty, they come with significant computational overhead. Single-model methods, while computationally favorable, often rely on heuristics and typically require additional propagation mechanisms for myopic uncertainty estimates. In this work we introduce universal value-function uncertainties (UVU), which, similar in spirit to random network distillation (RND), quantify uncertainty as squared prediction errors between an online learner and a fixed, randomly initialized target network. Unlike RND, UVU errors reflect policy-conditional *value uncertainty*, incorporating the future uncertainties *any given policy* may encounter. This is due to the training procedure employed in UVU: the online network is trained using temporal difference learning with a synthetic reward derived from the fixed, randomly initialized target network. We provide an extensive theoretical analysis of our approach using neural tangent kernel (NTK) theory and show that in the limit of infinite network width, UVU errors are exactly equivalent to the variance of an ensemble of independent universal value functions. Empirically, we show that UVU achieves equal performance to large ensembles on challenging multi-task offline RL settings, while offering simplicity and substantial computational savings.

## 1 Introduction

Deep reinforcement learning (RL) has emerged as an essential paradigm for addressing difficult sequential decision-making problems (Mnih et al., 2015; Silver et al., 2016; Vinyals et al., 2019) but a more widespread deployment of agents to real-world applications remains challenging. Open problems such as efficient exploration, scalable offline learning and safety pose persistent obstacles to this transition. Central to these capabilities is the quantification of *epistemic uncertainty*, an agent's uncertainty due to limited data. In the context of RL, uncertainty estimation relating to the *value function* is of particular importance as it reflects uncertainty about long-term consequences of actions.

However, computationally tractable estimation of value-function uncertainty remains a challenge. Bayesian RL approaches, both in its model-based (Ghavamzadeh et al., 2015) and model-free (Dearden et al., 1998) flavors, typically come with sound theoretical underpinnings but face significant computational hurdles due to the general intractability of posterior inference. Theoretical guarantees of the latter are moreover often complicated by the use of training procedures like temporal difference (TD) learning with bootstrapping. Conversely, deep ensembles (Lakshminarayanan et al., 2017) have emerged as a reliable standard for practical value uncertainty estimation in deep RL (Osband et al., 2016; Chen et al., 2017). Empirically, independently trained value functions from random initialization provide effective uncertainty estimates that correlate well with true estimation errors. Although in general more tractable than full posterior inference, this approach remains computationally challenging for larger models where a manyfold increase in computation and memory severely limits scalability. Various single-model approaches like random network distillation (RND) (Burda et al., 2019), pseudo counts (Bellemare et al., 2016) or intrinsic curiosity (Pathak et al., 2017) efficiently capture *myopic* epistemic uncertainty but require additional propagation mechanisms to obtain value

uncertainties (O'Donoghue et al., 2018; Janz et al., 2019; Zhou et al., 2020) and often elude a thorough theoretical understanding. We conclude that there persists a lack of computationally efficient single-model approaches with the ability to directly estimate policy-dependent value uncertainties with a strong theoretical foundation.

To this end, we introduce *universal value-function uncertainties* (UVU), a novel method designed to estimate epistemic uncertainty of value functions for any given policy using a single-model architecture. Similar in spirit to the well-known RND algorithm, UVU quantifies uncertainty through a prediction error between an online learner $u$ and a fixed, randomly initialized target network $g$. Crucially, and in contrast to the regression objective of RND, UVU optimizes its online network $u$ using temporal difference (TD) learning with a synthetic reward $r_g$ generated entirely from the target network $g$. By construction, the reward $r_g$ implies a value learning problem to which the target function $g$ itself is a solution, forcing the online learner $u$ to recover $g$ through minimization of TD losses. UVU then quantifies uncertainty as the squared prediction error between online learner and fixed target function. Unlike previous methods, our design requires no training of multiple models (e.g., ensembles) nor separate value and uncertainty models (e.g., RND, ICM). Furthermore, we design UVU as a universal policy-conditioned model (comparable to universal value function approximators (Schaul et al., 2015)), that is, it takes as input a state, action, and policy encoding and predicts the epistemic uncertainty associated with the value function for the encoded policy.

A key contribution of our work is a thorough theoretical analysis of UVU using the framework of neural tangent kernels (NTK) (Jacot et al., 2018). Specifically, we characterize the learning dynamics of wide neural networks with TD losses and gradient descent to obtain closed-form solutions for the convergence and generalization behavior of neural network value functions. In the limit of infinite network width, we then show that prediction errors generated by UVU are equivalent to the variance of an ensemble of universal value functions, both in expectation and with finite sample estimators.

We validate UVU empirically on an offline multi-task benchmark from the minigrid suite where agents are required to reject tasks they cannot perform to achieve maximal scores. We show that UVU's uncertainty estimates perform comparably to large deep ensembles, while drastically reducing the computational footprint.

## 2 Preliminaries

We frame our work within the standard Markov Decision Process (MDP) (Bellman, 1957) formalism, defined by the tuple $(\mathcal{S}, \mathcal{A}, \mathcal{R}, \gamma, P, \mu)$. Here, $\mathcal{S}$ is the state space, $\mathcal{A}$ is the action space, $\mathcal{R} : \mathcal{S} \times \mathcal{A} \to \mathscr{P}(\mathbb{R})$ is the distribution of immediate rewards, $\gamma \in [0, 1)$ is the discount factor, $P : \mathcal{S} \times \mathcal{A} \to \mathscr{P}(\mathcal{S})$ is the transition probability kernel, and $\mu : \mathscr{P}(\mathcal{S})$ is the initial state distribution. An RL agent interacts with this environment by selecting actions according to a policy $\pi : \mathcal{S} \to \mathscr{P}(\mathcal{A})$. At each timestep $t$, the agent is in state $S_t$, takes action $A_t \sim \pi(\cdot|S_t)$, receives a reward $R_t \sim \mathcal{R}(\cdot|S_t, A_t)$, and transitions to a new state $S_{t+1} \sim P(\cdot|S_t, A_t)$. We quantify the merit of taking actions $A_t = a$ in state $S_t = s$ and subsequently following policy $\pi$ by the action-value function, or Q-function $Q^\pi : \mathcal{S} \times \mathcal{A} \to \mathbb{R}$, which accounts for the cumulative discounted future rewards and adheres to a recursive consistency condition described by the Bellman equation

$$Q^\pi(s, a) = \mathbb{E}_{\mathcal{R}, \pi, P}[R_0 + \gamma Q^\pi(S_1, A_1)|S_0 = s, A_0 = a]. \tag{1}$$

The agent's objective then is to maximize expected returns $J(\pi) = \mathbb{E}_{S_0 \sim \mu, A_0 \sim \pi(\cdot|S_0)}[Q^\pi(S_0, A_0)]$.

Often, we may be interested in agents capable of operating a variety of policies to achieve different goals. Universal value function approximators (UVFAs) (Schaul et al., 2015) address this by conditioning value functions additionally on an encoding $z \in \mathcal{Z}$. This encoding specifies a current policy context, indicating for example a task or goal. We denote such *universal Q*-functions as $Q(s, a, z)$. In the context of this work, we consider $z$ to be a parameterization or indexing of a specific policy $\pi(\cdot|s, z)$, or in other words $Q : \mathcal{S} \times \mathcal{A} \times \mathcal{Z} \to \mathbb{R}$, $\quad Q(s, a, z) \equiv Q^{\pi(\cdot|s,z)}(s, a)$.

Both in the single and multi task settings, obtaining effective policies may require efficient exploration and an agent's ability to reason about *epistemic* uncertainty. This source of uncertainty, in contrast to *aleatoric* uncertainty, stems from a lack of knowledge and may in general be reduced by the acquisition of data. In the context of RL, we make an additional distinction between *myopic uncertainty* and *value uncertainty*.

## 2.1 Myopic Uncertainty and Neural Tangent Kernels

Myopic uncertainty estimation methods, such as RND or ensembles predicting immediate rewards or next states, quantify epistemic uncertainty without explicitly accounting for future uncertainties along trajectories. We first briefly recall the RND algorithm (Burda et al., 2019) , before introducing the neural tangent kernel (NTK) (Jacot et al., 2018) framework.

Random network distillation comprises two neural networks: A fixed, randomly initialized *target network* $g(x; \psi_0)$, and a *predictor network* $u(x; \vartheta_t)$. The online predictor $u(x; \vartheta_t)$ is trained via gradient descent to minimize a square loss between its own predictions and the target network's output on a set of data points $\mathcal{X} = \{x_i \in \mathbb{R}^{d_{\text{in}}}\}_{i=1}^{N_D}$. The RND prediction error at a test point $x$ then serves as an uncertainty or novelty signal. The loss and error function of RND are then given as

$$\mathcal{L}_{\text{rnd}}(\theta_t) = \tfrac{1}{2}(u(\mathcal{X}; \theta_t) - g(\mathcal{X}; \psi_0))^2, \quad \text{and} \quad \epsilon_{\text{rnd}}^2(x; \vartheta_t, \psi_0) = \tfrac{1}{2}(u(x; \vartheta_t) - g(x; \psi_0))^2. \quad (2)$$

This mechanism relies on the idea that the predictor network recovers the outputs of the target network only for datapoints contained in the dataset $x_i \in \mathcal{X}$, while a measurable error $\epsilon_{\text{rnd}}^2$ persists for out-of-distribution test samples $x_T \notin \mathcal{X}$, yielding a measure of epistemic uncertainty.

Next, we introduce the framework of neural tangent kernels, an analytical framework we intend to employ for the study of neural network and deep ensemble behavior. Consider a neural network $f(x, \theta_t) : \mathbb{R}^{n_{\text{in}}} \to \mathbb{R}$ with hidden layer widths $n_1, \ldots, n_L = n$ and inputs $x \in \mathbb{R}^{n_{\text{in}}}$, a dataset $\mathcal{X}$, and labels $\mathcal{Y} = \{y_i \in \mathbb{R}\}_{i=1}^{N_D}$. Inputs $x_i$ may, for example, be state-action tuples and labels $y_i$ may be rewards. The network parameters $\theta_0 \in \mathbb{R}^{n_p}$ are initialized randomly $\theta_0 \sim \mathcal{N}(0, 1)$ and updated with gradient descent with infinitesimal step sizes, also called gradient flow. In the limit of infinite width $n$, the function initialization $f(\cdot, \theta_0)$, as shown by Lee et al. (2018), is equivalent to a Gaussian process prior with a specific kernel $\kappa : \mathbb{R}^{n_{\text{in}}} \times \mathbb{R}^{n_{\text{in}}} \to \mathbb{R}$ called the neural network Gaussian process (NNGP). The functional evolution of $f$ through gradient flow is then governed by a *gradient* inner product kernel $\Theta : \mathbb{R}^{n_{\text{in}}} \times \mathbb{R}^{n_{\text{in}}} \to \mathbb{R}$ yielding

$$\Theta(x, x') = \nabla_\theta f(x, \theta_0)^\top \nabla_\theta f(x', \theta_0), \quad \text{and} \quad \kappa(x, x') = \mathbb{E}[f(x, \theta_0) f(x', \theta_0)]. \quad (3)$$

Remarkably, seminal work by Jacot et al. (2018) showed that in the limit of infinite width and appropriate parametrization[1], the kernel $\Theta$ becomes deterministic and remains constant throughout training. This limiting kernel, referred to as the neural tangent kernel (NTK), leads to analytically tractable training dynamics for various loss functions, including the squared loss $\mathcal{L}(\theta_t) = \tfrac{1}{2}\|f(\mathcal{X}; \theta_t) - \mathcal{Y}\|_2^2$. Owing to this, one can show (Jacot et al., 2018; Lee et al., 2020) that for $t \to \infty$ post convergence function evaluations $f(\mathcal{X}_T, \theta_\infty)$ on a set of test points $\mathcal{X}_T$, too, are Gaussian with mean $\mathbb{E}[f(\mathcal{X}_T, \theta_\infty)] = \Theta_{\mathcal{X}_T \mathcal{X}} \Theta_{\mathcal{X}\mathcal{X}}^{-1} \mathcal{Y}$ and covariance

$$\text{Cov}[f(\mathcal{X}_T, \theta_\infty)] = \kappa_{\mathcal{X}_T \mathcal{X}_T} - \left(\Theta_{\mathcal{X}_T, \mathcal{X}} \Theta_{\mathcal{X}\mathcal{X}}^{-1} \kappa_{\mathcal{X}\mathcal{X}_T} + h.c.\right) + \Theta_{\mathcal{X}_T \mathcal{X}} \Theta_{\mathcal{X}\mathcal{X}}^{-1} \kappa_{\mathcal{X}\mathcal{X}} \Theta_{\mathcal{X}\mathcal{X}}^{-1} \Theta_{\mathcal{X}\mathcal{X}_T}, \quad (4)$$

where $h.c.$ denotes the Hermitian conjugate of the preceding term and we used the shorthands $\Theta_{\mathcal{X}_1 \mathcal{X}_2} = \Theta(\mathcal{X}_1, \mathcal{X}_2)$ and $\kappa_{\mathcal{X}_1 \mathcal{X}_2} = \kappa(\mathcal{X}_1, \mathcal{X}_2)$. This expression provides a closed-form solution for the epistemic uncertainty captured by an infinite ensemble of NNs in the NTK regime trained with square losses. For example, the predictive variances of such ensembles are easily obtained as the diagonal entries of Eq. 4. While requiring an idealized setting, NTK theory offers a solid theoretical grounding for quantifying the behavior of deep ensembles and, by extension, myopic uncertainty estimates from related approaches. However, this analysis does not extend to value functions trained with TD losses and bootstrapping as is common in practical reinforcement learning settings.

## 2.2 Value Uncertainty

In contrast to myopic uncertainties, value uncertainty quantifies a model's lack of knowledge in the value $Q^\pi(s, a)$. As such it inherently depends on future trajectories induced by policies $\pi$. Due to this need to account for accumulated uncertainties over potentially long horizons, value uncertainty estimation typically renders more difficult than its myopic counterpart.

A widely used technique(Osband et al., 2016; Chen et al., 2017; An et al., 2021) to this end is the use of deep ensembles of value functions $Q(s, a, \theta_t) : \mathcal{S} \times \mathcal{A} \to \mathbb{R}$ from random initializations $\theta_0$. Q-functions are trained on transitional data $\mathcal{X}_{TD} = \{s_i, a_i\}_{i=1}^{N_D}$, $\mathcal{X}'_{TD} = \{s'_i, a'_i\}_{i=1}^{N_D}$, and $r = \{r_i\}_{i=1}^{N_D}$,

---

[1]so-called NTK parametrization scales forward/backward passes appropriately, see Jacot et al. (2018)

where $s_i'$ are samples from the transition kernel $P$ and $a_i'$ are samples from a policy $\pi$. $Q$-functions are then optimized through gradient descent on a temporal difference (TD) loss given by

$$\mathcal{L}(\theta_t) = \tfrac{1}{2} \| \left[ \gamma Q^\pi(\mathcal{X}'_{TD}, \theta_t) \right]_{\text{sg}} + r - Q^\pi(\mathcal{X}_{TD}, \theta_t) \|_2^2, \tag{5}$$

where $[\cdot]_{\text{sg}}$ indicates a stop-gradient operation. Due to the stopping of gradient flow through $Q(\mathcal{X}', \theta_t)$, we refer to this operation as semi-gradient updates. Uncertainty estimates can then be obtained as the variance $\sigma_q^2(s, a) = \mathbb{V}_{\theta_0}[Q(s, a, \theta_t)]$ between ensembles of $Q$-functions from random initializations. While empirically successful, TD-trained deep ensembles are not as well understood as the supervised learning setting outlined in the previous section 2.1. Due to the use of bootstrapped TD losses, the closed-form NTK regime solutions in Eq. 4 do not apply to deep value function ensembles.

An alternative to the above approach is the propagation of myopic uncertainty estimates. Several prior methods(O'Donoghue et al., 2018; Zhou et al., 2020; Luis et al., 2023) formalize this setting under a model-based perspective, where transition models $\tilde{P}(\cdot|s, a)$ are sampled from a Bayesian posterior conditioned on transition data up to $t$. For acyclic MDPs, this setting permits a consistency condition similar to the Bellman equation that upper bounds value uncertainties recursively. While this approach devises a method for obtaining value uncertainties from propagated myopic uncertainties, several open problems remain, such as the tightness of model-free bounds of this kind (Janz et al., 2019; Van der Vaart et al., 2025) as well as how to prevent *underestimation* of these upper bounds due to the use of function approximation (Rashid et al., 2020; Zanger et al., 2024).

## 3 Universal Value-Function Uncertainties

Our method, *universal value-function uncertainties* (UVU), measures epistemic value uncertainty as the prediction errors between an online learner and a fixed target network, similar in spirit to random network distillation (Burda et al., 2019). However, while RND quantifies myopic uncertainty through immediate prediction errors, UVU modifies the training process of the online learner such that the resulting prediction errors reflect value-function uncertainties, that is, uncertainty about long-term returns under a given policy.

Our method centers around the interplay of two distinct neural networks: an online learner $u(s, a, z, \vartheta_t) : \mathcal{S} \times \mathcal{A} \times \mathcal{Z} \to \mathbb{R}$, parameterized by weights $\vartheta_t$, and a fixed, randomly initialized target network $g(s, a, z, \psi_0) : \mathcal{S} \times \mathcal{A} \times \mathcal{Z} \to \mathbb{R}$, parameterized by weights $\psi_0$. Given a transition $(s, a, s')$ and policy encoding $z$, we draw subsequent actions $a'$ from a policy $\pi(\cdot|s', z)$. Then, we use the fixed target network $g$ to generate synthetic rewards as

$$r_g^z(s, a, s', a') = g(s, a, z, \psi_0) - \gamma g(s', a', z, \psi_0). \tag{6}$$

While the weights $\psi_0$ of the target network remain fixed at initialization, the online network $u$ is trained to minimize a TD loss using the synthetic reward $r_g^\pi$. Given a dataset $\mathcal{X} = \{s_i, a_i, z_i\}_{i=1}^{N_D}$, we have

$$\mathcal{L}(\vartheta_t) = \frac{1}{2N_D} \sum_i^{N_D} \left( \gamma \left[ u(s_i', a_i', z_i, \vartheta_t) \right]_{\text{sg}} + r_g^z(s_i, a_i, s_i', a_i') - u(s_i, a_i, z_i, \vartheta_t) \right)^2, \tag{7}$$

where $[\cdot]_{\text{sg}}$ indicates a stop-gradient operation. For any tuple $(s, a, z)$ ($\in \mathcal{X}$ or not), we measure predictive uncertainties as squared prediction errors between the learner and the target function

$$\epsilon(s, a, z, \vartheta_t, \psi_0)^2 = \left( u(s, a, z, \vartheta_t) - g(s, a, z, \psi_0) \right)^2. \tag{8}$$

The intuition behind this design is that, by construction, the value-function associated with policy $\pi(\cdot|s, z)$ and the synthetic rewards $r_g^z(s, a, s', a')$ exactly equals the fixed target network $g(s, a, z, \psi_0)$. As a sanity check, note that the target function $g(s, a, z, \psi_0)$ itself satisfies the Bellman equation for the policy $\pi(\cdot|s, z)$ and the synthetic reward definition in Eq. (6), constituting a *random value function* to $r_g^z$ and hence achieves zero-loss according to Eq. (7). Therefore, if the dataset $\mathcal{X}$ sufficiently covers the dynamics induced by $\pi(\cdot|s, z)$, the online network $u(s, a, z, \vartheta_0)$ is able to recover $g(s, a, z, \psi_0)$ exactly, nullifying prediction errors. However, when data coverage is incomplete for the evaluated policy, minimization of the TD loss 7 is not sufficient for the online network $u(s, a, z, \vartheta_0)$ to recover target network predictions $g(s, a, z, \psi_0)$. This discrepancy is captured by the prediction errors, which quantify epistemic uncertainty regarding future gaps of the available data.

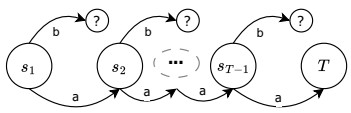

Figure 2: (*left*:) Illustration of uncertainty estimation in tabular UVU with 4 independently initialized tables for $u$ and $g$. Access to full trajectory data allows $u$ to recover $g$. (*right*:) By executing action "b", trajectories are effectively truncated, preventing $u$ from recovering $g$. All plots use $\gamma = 0.7$.

## 3.1 BUILDING INTUITION BY AN EXAMPLE

To build intuition for how UVU operates and captures value uncertainty, we first consider a tabular setting with a simple chain MDP as illustrated in Figure 1. Suppose we collect data from a deterministic policy $\pi_d$ using action $a$ exclusively. Given this dataset, suppose we would like to estimate the uncertainty associated with the value $Q^{\pi(\cdot|s,z)}(s,a)$ of a policy $\pi(\cdot|s,z)$ that differs from the data-collection policy in that it chooses action "$b$" in $s_3$. In our tabular setting, we then initialize random tables $u_{sa}$ and $g_{sa}$. For every transition $(s_t, a_t, s_{t+1})$ contained in our single-trajectory dataset, we draw $a_{t+1} \sim \pi(\cdot|s,z)$, compute the reward $r_{g,t}$ as $r_{g,t} = g_{s_t a_t} - \gamma g_{s_{t+1} a_{t+1}}$ and update table entries with the rule $u_{s_t a_t} \leftarrow r_{g,t} + \gamma u_{s_{t+1} a_{t+1}}$. Fig. 2 visualizes this process for several independently initialized tables (rows in Fig. 2) for the data-collecting policy $\pi_d$ (left), and for the altered policy $\pi(\cdot|s,z)$ (right), which chooses action "$b$" in $s_3$. We outline how this procedure yields uncertainty estimates: We first note,

Figure 1: Chain MDP of length $N$ with unexplored actions $b$.

that one may regard $g$ as a randomly generated value-function, for which we derive the corresponding reward function as $r_g$. As $g_{sa}$, by construction, is the value-function corresponding to $r_g$, one may expect that the update rule applied to $u_{sa}$ causes $u_{sa}$ to recover $g_{sa}$. Crucially, however, this is only possible if sufficient data is available for the evaluated policy. When a policy diverges from available data, as occurs under $\pi(\cdot|s,z)$ in $s_3$, this causes an effective truncation of the collected trajectory. Consequently, $u_{s_1 a}$ and $u_{s_2 a}$ receive updates from $u_{s_3 b}$, which remains at its initialization, rather than inferring the reward-generating function $g_{sa}$. In the absence of long-term data, the empirical Bellman equations reflected in our updates do not uniquely determine the underlying value function $g_{sa}$. Indeed, both $u_{sa}$ and $g_{sa}$ incur zero TD-error in the r.h.s. of Fig. 2, yet differ significantly from each other. It is this ambiguity that UVU errors $(g_{sa} - u_{sa})^2$ quantify. To ensure $u$ recovers $g$, longer rollouts under the policy $\pi(\cdot|s,z)$ are required to sufficiently constrain the solution space dictated by the Bellman equations (as seen in Fig. 2 left).

Figure 3 illustrates uncertainty estimates for the shown chain MDP using neural networks and for a whole family of policies $\pi(\cdot|s,z)$ which select the unexplored action $b$ with probability $1 - z$. We analyze the predictive variance of an ensemble of 128 universal $Q$-functions, each conditioned on the policy $\pi(\cdot|s,z)$. In the bottom row, we plot the squared prediction error of a single UVU model, averaged over 128 independent heads. Both approaches show peaked uncertainty in early sections, as policies are more likely to choose the unknown action "$b$" eventually, and low uncertainty closer to the terminal state and for $z$ close to 1. A comparison with RND is provided in the Appendix B.3.

## 4 WHAT UNCERTAINTIES DO UNIVERSAL VALUE-FUNCTION UNCERTAINTIES LEARN?

While the previous section provided intuition for UVU, we now derive an analytical characterization of the uncertainties captured by the prediction errors $\epsilon$ between a converged online learner $u$ and the fixed target $g$. We turn to NTK theory to characterize the generalization properties of the involved neural networks in the limit of infinite width, allowing us to draw an exact equality between the squared predictions errors of UVU and the variance of universal value function ensembles.

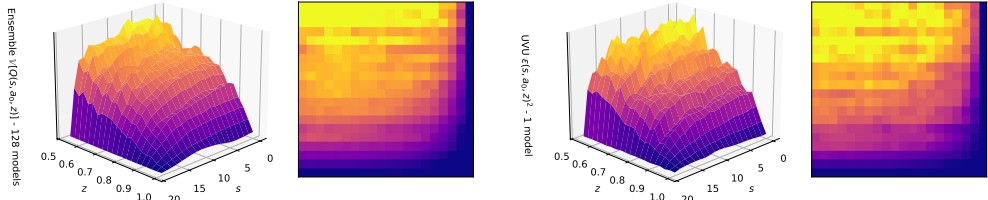

Figure 3: *From left to right, (1. and 2.):* Variance of an ensemble of 128 universal Q-functions trained on a chain MDP dataset. *(3. and 4.):* Value uncertainty as measured by UVU prediction errors with a single 128-headed model. All plots evaluate the "$a$" action of the chain MDP.

In the following analysis, we use the notational shorthand $x = (s, a, z)$ and $x' = (s', a', z)$ and denote a neural network $f(x, \theta_t)$ with hidden layer widths $n_1, \ldots, n_L = n$, transitions from $\mathcal{X} = \{(s_i, a_i, z_i)\}_{i=1}^{N_D}$ to $\mathcal{X}' = \{(s'_i, a'_i, z_i)\}_{i=1}^{N_D}$, where $a'_i \sim \pi(\cdot|s'_i, z_i)$, and rewards $r = \{r_i\}_{i=1}^{N_D}$. The evolution of the parameters $\theta_t$ under gradient descent with infinitesimal step sizes, also called gradient flow, is driven by the minimization of TD losses with

$$\frac{\mathrm{d}}{\mathrm{d}t}\theta_t = -\alpha\nabla_\theta\mathcal{L}(\theta_t), \quad \text{and} \quad \mathcal{L}(\theta_t) = \tfrac{1}{2}\| [\gamma f(\mathcal{X}', \theta_t)]_{sg} + r - f(\mathcal{X}, \theta_t) \|_2^2. \quad (9)$$

We study the dynamics induced by this parameter evolution in the infinite-width limit $n \to \infty$. In this regime, the learning dynamics of $f$ become linear as the NTK becomes deterministic and stationary, permitting explicit closed-form expressions for the evolution of the function $f(x, \theta_t)$. In particular, we show that the post convergence function $\lim_{t\to\infty} f(x, \theta_t)$ is given by

$$f(x, \theta_\infty) = f(x, \theta_0) - \Theta_{x\mathcal{X}}(\Theta_{\mathcal{X}\mathcal{X}} - \gamma\Theta_{\mathcal{X}'\mathcal{X}})^{-1}(f(\mathcal{X}, \theta_0) - (\gamma f(\mathcal{X}', \theta_0) + r)), \quad (10)$$

where $\Theta_{xx'}$ is the NTK of $f$. Proof is given in Appendix A.1. This identity is useful to our analysis as it delineates any converged function $f(x, \theta_\infty)$ trained with TD losses 9 through its initialization $f(x, \theta_0)$. Theorem 1 leverages this deterministic dependency to express the distribution of post convergence functions over random initializations $\theta_0$.

**Theorem 1.** *Let $f(x, \theta_t)$ be a NN with $L$ hidden layers of width $n_1, \ldots, n_L = n$ trained with gradient flow to reduce the TD loss $\mathcal{L}(\theta_t) = \frac{1}{2}\| \gamma[f(\mathcal{X}', \theta_t)]_{sg} + r - f(\mathcal{X}, \theta_t) \|_2^2$. In the limit of infinite width $n \to \infty$ and time $t \to \infty$, the distribution of predictions $f(\mathcal{X}_T, \theta_\infty)$ on a set of test points $\mathcal{X}_T$ converges to a Gaussian with mean and covariance given by*

$$\mathbb{E}_{\theta_0}\big[f(\mathcal{X}_T, \theta_\infty)\big] = \Theta_{\mathcal{X}_T\mathcal{X}}\Delta_{\mathcal{X}}^{-1}r,$$

$$\mathrm{Cov}_{\theta_0}\big[f(\mathcal{X}_T, \theta_\infty)\big] = \kappa_{\mathcal{X}_T\mathcal{X}_T} - (\Theta_{\mathcal{X}_T\mathcal{X}}\Delta_{\mathcal{X}}^{-1}\Lambda_{\mathcal{X}_T} + h.c.) + (\Theta_{\mathcal{X}_T\mathcal{X}}\Delta_{\mathcal{X}}^{-1}(\Lambda_{\mathcal{X}} - \gamma\Lambda_{\mathcal{X}'})\Delta_{\mathcal{X}}^{-1\top}\Theta_{\mathcal{X}\mathcal{X}_T}),$$

*where $\Theta_{xx'}$ is the NTK, $\kappa_{xx'}$ is the NNGP kernel, h.c. denotes the Hermitian conjugate, and*

$$\Delta_{\tilde{\mathcal{X}}} = \Theta_{\mathcal{X}\tilde{\mathcal{X}}} - \gamma\Theta_{\mathcal{X}'\tilde{\mathcal{X}}}, \quad \text{and} \quad \Lambda_{\tilde{\mathcal{X}}} = \kappa_{\mathcal{X}\tilde{\mathcal{X}}} - \gamma\kappa_{\mathcal{X}'\tilde{\mathcal{X}}}.$$

Proof is provided in Appendix A.1. Theorem 1 is significant as it allows us to formalize explicitly the expected behavior and uncertainties of neural networks trained with semi-gradient TD losses, including universal value function ensembles and the prediction errors of UVU. In particular, the variance of an ensemble of universal $Q$-functions $Q(\mathcal{X}_T, \theta_\infty)$ over random initializations $\theta_0$ is readily given by the diagonal entries of the covariance matrix $\mathrm{Cov}[Q(\mathcal{X}_T, \theta_\infty)]$. Applied to the UVU setting, Theorem 1 gives an expression for the converged online network $u(x, \vartheta_\infty) = \Theta_{x\mathcal{X}}\Delta_{\mathcal{X}}^{-1}r_g^z$ trained with the synthetic rewards $r_g^z = g(\mathcal{X}, \psi_0) - \gamma g(\mathcal{X}', \psi_0)$. From this, It is straightforward to obtain the distribution of post convergence prediction errors $\frac{1}{2}\epsilon(x, \vartheta_\infty, \psi_0)^2$. In Corollary 1, we use this insight to conclude that the expected squared prediction errors of UVU precisely match the variance of value functions $Q(x, \theta_\infty)$ from random initializations $\theta_0$.

**Corollary 1.** *Under the conditions of Theorem 1, let $u(x, \vartheta_\infty)$ be a converged online predictor trained with synthetic rewards generated by the fixed target network $g(x, \psi_0)$ with $r_g^z = g(\mathcal{X}, \psi_0) - \gamma g(\mathcal{X}', \psi_0)$. Furthermore denote the variance of converged universal Q-functions $\mathbb{V}_{\theta_0}[Q(x, \theta_\infty)]$.*

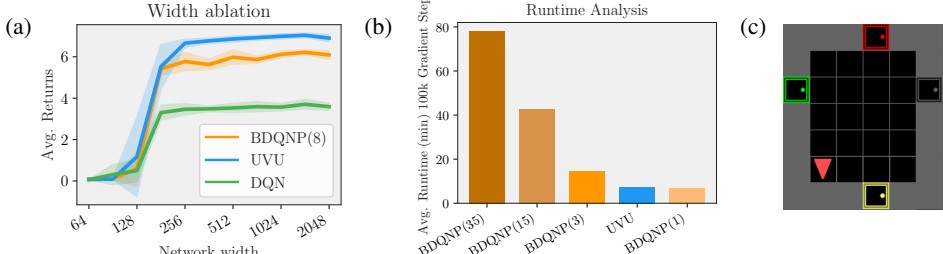

Figure 4: (a) Ablation on `GoToDoor`-10 with different network widths. Shaded region indicates standard deviations over 5 seeds. (b) Runtime of various ensemble sizes vs. UVU. Ensembles are implemented with `vmap` in JAX(Bradbury et al., 2018). (c) Illustration of the `GoToDoor` environment. The agent (red triangle) must navigate to the door indicated by the task specification $z$.

*Assume $u$, $g$, and $Q$ are architecturally equal and parameters are drawn i.i.d. $\theta_0, \vartheta_0, \psi_0 \sim \mathcal{N}(0, 1)$. The expected squared prediction error coincides with $Q$-function variance*

$$\mathbb{E}_{\vartheta_0, \psi_0}\left[\tfrac{1}{2}\epsilon(x, \vartheta_\infty, \psi_0)^2\right] = \mathbb{V}_{\theta_0}\left[Q(x, \theta_\infty)\right], \tag{11}$$

*where the l.h.s. expectation and r.h.s. variance are taken over random initializations $\vartheta_0, \psi_0, \theta_0$.*

Proof is given in Appendix A.1.3. This result provides the central theoretical justification for UVU: in the limit of infinite width, our measure of uncertainty, the expected squared prediction error between the online and target network, is mathematically equivalent to the variance one would obtain by training an ensemble of universal $Q$-functions.

In practice, we are moreover interested in the behavior of finite estimators, that is, ensemble variances are estimated with a finite number of models. We furthermore implement UVU with a number of multiple independent heads $u_i$ and $g_i$ with shared hidden layers. Corollary 2 shows that the distribution of the sample mean squared prediction error from $M$ heads is identical to the distribution of the sample variance of an ensemble of $M + 1$ independently trained universal $Q$-functions.

**Corollary 2.** *Under the conditions of Theorem 1, consider online and target networks with $M$ independent heads $u_i, g_i$, $i = 1, \ldots, M$, each trained to convergence with errors $\epsilon_i(x, \vartheta_\infty, \psi_0)$. Let $\tfrac{1}{2}\bar{\epsilon}(x, \vartheta_\infty, \psi_0)^2 = \tfrac{1}{2M}\sum_{i=1}^{M} \epsilon_i(x, \vartheta_\infty, \psi_0)^2$ be the sample mean squared prediction error over $M$ heads. Moreover, consider $M + 1$ independent converged $Q$-functions $Q_i(x; \theta_\infty)$ and denote their sample variance $\bar{\sigma}_Q^2(x, \theta_\infty) = \tfrac{1}{M}\sum_{i=1}^{M+1}(Q_i(x; \theta_\infty) - \bar{Q}(x; \theta_\infty))^2$, where $\bar{Q}$ is the sample mean. The two estimators are identically distributed according to a scaled Chi-squared distribution*

$$\tfrac{1}{2}\bar{\epsilon}(x, \vartheta_\infty, \psi_0)^2 \stackrel{D}{=} \bar{\sigma}_Q^2(x, \theta_\infty), \quad \bar{\sigma}_Q^2(x, \theta_\infty) \sim \frac{\sigma_Q^2}{M}\chi^2(M), \tag{12}$$

*with $M$ degrees of freedom and $\sigma_Q^2(x, \theta_\infty) = \mathbb{V}_{\theta_0}[Q(x, \theta_\infty)]$ is the analytical variance of converged $Q$-functions given by Theorem 1.*

Proof is provided in Appendix A.2.3. The distributional equivalence of these finite sample estimators provides theoretical motivation for using a multi headed architecture with shared hidden layers within a single UVU model and its use as an estimator for ensemble variances of universal $Q$-functions. While the assumptions of infinite width and gradient flow are theoretical idealizations, several empirical results suggest that insights from the NTK regime can translate well to practical finite width deep learning models (Lee et al., 2020; Liu et al., 2020; Tsilivis and Kempe, 2022), motivating further empirical investigation in Section 5.

## 5 EMPIRICAL ANALYSIS

Our empirical analysis is designed to assess whether UVU can effectively quantify value function uncertainty in practical settings, comparing its performance against established baselines, particularly deep ensembles. Specifically, we aim to address the following questions:

1. Does the theoretical motivation for UVU hold in practice and do its uncertainty estimates enable effective decision-making comparable to deep ensembles?

2. How are uncertainty estimates generated by UVU affected by deviations from our theoretical analysis, namely finite network width?

To address these questions, we focus on an offline multitask RL setting with incomplete data where reliable uncertainty estimation is crucial to attain high performance.

Code for experimental reproduction is available at
`github.com/anyboby/universal-value-function-uncertainties` .

## 5.1 EXPERIMENTAL SETUP

In our experimental analysis, we use an offline variant of the `GoToDoor` environment from the Minigrid benchmark suite (Chevalier-Boisvert et al., 2023). An example view is shown in Figure 4 (c). In this task, the agent navigates a grid world containing four doors of different colors, placed at random locations and receives a task specification $z$ indicating a target door color. Upon opening the correct door, the agent receives a reward and is placed in a random different location. Episodes are of fixed length and feature a randomly generated grid layout and random door positions / colors. In our experiments, we use variations of different difficulties by increasing maximum grid sizes.

**Dataset Collection.** A dataset $\mathcal{D} = \{(s_i, a_i, r_i, z_i, s'_i,)\}_{i=1}^{N_D}$ is collected using a policy that performs expertly but systematically fails for certain task/grid combinations (e.g., it can not successfully open doors on the "north" wall, irrespective of color or grid layout). Policies seeking to improve upon the behavior policy thus ought to deviate from the dataset, inducing value uncertainty.

**Task Rejection Protocol.** All baselines implement a DQN-based agent trained in an offline fashion on $\mathcal{D}$. As the agents aim to learn an optimal policy for all grids and tasks contained in $\mathcal{D}$, the resulting greedy policy tends to deviate from the available data when the collecting policy is suboptimal. We employ a task-rejection protocol to quantify an agent's ability to recognize this divergence and the associated value uncertainty. As most task/grid combinations are contained in $\mathcal{D}$, though with varying levels of policy expertise, myopic uncertainty is not sufficient for fulfilling this task. Specifically, upon encountering the initial state $s_0$, the agent is given opportunity to reject a fixed selection of tasks (here door colors). It is subsequently given one of the remaining, non-rejected tasks and performance is measured by the average return achieved on the attempted task. Successful agents must thus either possess uncertainty estimates reliable enough to consistently reject tasks associated with a data/policy mismatch or rely on out-of-distribution generalization. Similar protocols, known as accuracy rejection curves, have been used widely in the supervised learning literature(Nadeem et al., 2009).

## 5.2 RESULTS

We conduct experiments according to the above protocol and perform a quantitative evaluation of UVU and several baseline algorithms. All agents are trained offline and use the basic DQN architecture (Mnih et al., 2015) adapted for universal value functions, taking the task encoding $z$ as an additional input to the state (details are provided in Appendix B). Specifically, we compare UVU against several baselines: A DQN baseline with random task rejection (DQN); Bootstrapped DQN with randomized priors (BDQNP) (Osband et al., 2019); A DQN adaptation of random network distillation (DQN-RND) (Burda et al., 2019) and a version adapted with the uncertainty prior mechanism proposed by Zanger et al. (2024) (DQN-RND-P). Except for the DQN baseline, all algorithms reject tasks based on the highest uncertainty estimate, given the initial state $s_0$ and action $a_0$, which is chosen greedily by the agent.

Table 1 shows the average return achieved by each method on the GoToDoor experiment across different maximum grid sizes, with average runtimes displayed in Fig. 4 (b). This result addresses our first research question regarding the practical effectiveness of UVU compared to ensembles and other baseline methods. As shown, the standard DQN baseline performs significantly worse than uncertainty-based algorithms, indicating that learned $Q$-functions do not generalize sufficiently to counterbalance inadequate uncertainty estimation. Both small and large ensembles significantly improve performance by leveraging uncertainty to reject tasks and policies associated with missing

Table 1: Results of offline multitask RL with task rejection on different variations of the `GoToDoor` environment. Results are average evaluation returns of the best-performing policy over $10^5$ gradient steps and intervals are $90\%$ student's $t$ confidence intervals from 10 independent seeds.

| Size | DQN | BDQNP(3) | BDQNP(15) | BDQNP(35) | DQN-RND | DQN-RND-P | UVU (Ours) |
|---|---|---|---|---|---|---|---|
| 5 | $5.50 \pm .15$ | $8.69 \pm .24$ | $\mathbf{10.50 \pm .04}$ | $\mathbf{10.58 \pm .03}$ | $3.94 \pm .50$ | $\mathbf{10.41 \pm .12}$ | $\mathbf{10.54 \pm .03}$ |
| 6 | $4.93 \pm .12$ | $7.66 \pm .09$ | $9.39 \pm .04$ | $\mathbf{9.57 \pm .04}$ | $1.99 \pm .40$ | $9.28 \pm .12$ | $\mathbf{9.54 \pm .03}$ |
| 7 | $4.58 \pm .09$ | $6.61 \pm .16$ | $8.49 \pm .05$ | $\mathbf{8.75 \pm .06}$ | $2.66 \pm .43$ | $8.12 \pm .23$ | $\mathbf{8.73 \pm .04}$ |
| 8 | $4.06 \pm .12$ | $5.91 \pm .10$ | $7.68 \pm .05$ | $7.92 \pm .05$ | $2.53 \pm .54$ | $7.40 \pm .14$ | $\mathbf{8.03 \pm .04}$ |
| 9 | $3.66 \pm .09$ | $5.04 \pm .08$ | $6.69 \pm .07$ | $7.03 \pm .13$ | $2.39 \pm .38$ | $6.39 \pm .19$ | $\mathbf{7.29 \pm .10}$ |
| 10 | $3.39 \pm .11$ | $4.64 \pm .14$ | $6.09 \pm .13$ | $\mathbf{6.53 \pm .16}$ | $2.25 \pm .48$ | $5.64 \pm .17$ | $\mathbf{6.72 \pm .12}$ |

data. RND-based agents perform well when intrinsic reward priors are used. Our approach scores highly and outperforms many of the tested baselines with statistical significance, indicating that it is indeed able to effectively quantify value uncertainty using a single-model multi-headed architecture.

We furthermore ablate UVU's dependency on network width, given that our theoretical analysis is situated in the infinite width limit. Fig. 4 (a) shows that UVU's performance scales similarly with network width to DQN and BDQNP baselines, indicating that finite-sized networks, provided appropriate representational capacity, are sufficient for effective uncertainty estimates.

## 6 RELATED WORK

A body of literature considers the quantification of value function uncertainty in the context of exploration. Early works (Dearden et al., 1998; Engel et al., 2005) consider Bayesian adoptions of model-free RL algorithms. More recent works provide theoretical analyses of the Bayesian model-free setting and correct applications thereof (Fellows et al., 2021; Schmitt et al., 2023; Van der Vaart et al., 2025), which is a subject of debate due to the use TD losses. Several works furthermore derive provably efficient model-free algorithms using frequentist upper bounds on values in tabular (Strehl et al., 2006; Jin et al., 2018) and linear settings (Jin et al., 2020). Similarly, Yang et al. (2020) derive provably optimisic bounds of value functions in the NTK regime, but in contrast to our work uses local bonuses to obtain these. The exact relationship between bounds derived from local bonuses and the functional variance in ensemble or Bayesian settings remains open.

The widespread use and empirical success of ensembles for uncertainty quantification in deep learning (Dietterich, 2000; Lakshminarayanan et al., 2017) has motivated several directions of research towards a better theoretical understanding of their behavior. Following seminal works by Jacot et al. (2018) and Lee et al. (2020) who characterize NN learning dynamics in the NTK regime, a number of works have connected deep ensembles to Bayesian interpretations (He et al., 2020; D'Angelo and Fortuin, 2021). Moreover, a number of papers have studied the learning dynamics of model-free RL: in the overparametrized linear settings (Xiao et al., 2021); in neural settings for single (Cai et al., 2019) and multiple layers (Wai et al., 2020); to analyze generalization behavior (Lyle et al., 2022) with linear and second-order approximations. It should be noted that the aforementioned do not focus on probabilistic descriptions of posterior distributions in the NTK regime. In contrast, our work provides probabilistic closed-form solutions for this setting with semi-gradient TD learning. In practice, the use of deep ensembles is common in RL, with applications ranging from efficient exploration (Osband et al., 2016; Chen et al., 2017; Osband et al., 2019; Nikolov et al., 2019; Zanger et al., 2024) to off-policy or offline RL (An et al., 2021; Chen et al., 2021; Lee et al., 2021) and conservative or safe RL (Lütjens et al., 2019; Lee et al., 2022; Hoel et al., 2023). Single model methods that aim to reduce the computational burden of ensemble methods typically operate as myopic uncertainty estimators (Burda et al., 2019; Pathak et al., 2017; Lahlou et al., 2021; Zanger et al., 2025) and require additional mechanisms (O'Donoghue et al., 2018; Janz et al., 2019; Zhou et al., 2020; Luis et al., 2023).

## 7 LIMITATIONS AND DISCUSSION

In this work, we introduced universal value-function uncertainties (UVU), an efficient single-model method for uncertainty quantification in value functions. Our method measures uncertainties as prediction error between a fixed, random target network and an online learner trained with a temporal difference (TD) loss. This induces prediction errors that reflect long-term, policy-dependent

uncertainty rather than myopic novelty. One of our core contributions is a thorough theoretical analysis of this approach via neural tangent kernel theory, which, in the limit of infinite network width, establishes an equivalence between UVU errors and the variance of ensembles of universal value functions. Empirically, UVU achieves performance comparable and sometimes superior to sizeable deep ensembles and other baselines in challenging offline task-rejection settings, while offering substantial computational savings.

We believe our work opens up several avenues for future research: Although our NTK analysis provides a strong theoretical backing, it relies on idealized assumptions, notably the limit of infinite network width (a thoroughgoing exposition of our approximations is provided in Appendix A.3). Our experiments suggest UVU's performance is robust in practical finite-width regimes (Figure 4), yet bridging this gap between theory and practice remains an area for future work. On a related note, analysis in the NTK regime typically eludes feature learning. Combinations of UVU with representation learning approaches such as self-predictive auxiliary losses (Schwarzer et al., 2020; Guo et al., 2022; Fujimoto et al., 2023) are, in our view, a very promising avenue for highly challenging exploration problems. Furthermore, while our approach estimates uncertainty for given policies, it does not devise a method for obtaining diverse policies and encodings thereof. We thus believe algorithms from the unsupervised RL literature(Touati and Ollivier, 2021; Zheng et al., 2023) naturally integrate with our approach. In conclusion, we believe UVU provides a strong foundation for future developments in uncertainty-aware agents that are both capable and computationally feasible.

## 8 ACKNOWLEDGEMENTS

This project has received funding from the EU Horizon 2020 programme *Epistemic AI* under grant number 964505 and the Dutch Research Council (NWO) project *Reliable Out-of-Distribution Generalization in Deep Reinforcement Learning* with project number OCENW.M.21.234. Computational resources for experimental studies were provided by the Delft High Performance Computing Centre (DHPC) and the Delft Artificial Intelligence Cluster  (DAIC).

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

# A  THEORETICAL RESULTS

This section provides proofs and further theoretical results for universal value-function uncertainties (UVU).

## A.1  LEARNING DYNAMICS OF UVU

We begin by deriving learning dynamics for general functions with temporal difference (TD) losses and gradient descent, before analyzing the post training distribution of deep ensembles and prediction errors of UVU.

### A.1.1  LINEARIZED LEARNING DYNAMICS WITH TEMPORAL DIFFERENCE LOSSES

We analyze the learning dynamics of a function trained using semi-gradient temporal difference (TD) losses on a fixed dataset of transitions $\mathcal{X}, \mathcal{X}'$. Let $f(x, \theta_t)$ denote a NN of interest with depth $L$ and widths $n_1, \ldots, n_{L-1} = n$.

**Proposition 1.** *In the limit of infinite width $n \to \infty$ and infinite time $t \to \infty$, the function $f(x, \theta_t)$ converges to*

$$f(x, \theta_\infty) = f(x, \theta_0) - \Theta_{x\mathcal{X}} \left(\Theta_{\mathcal{X}\mathcal{X}} - \gamma\Theta_{\mathcal{X}'\mathcal{X}}\right)^{-1} \left(f(\mathcal{X}, \theta_0) - (\gamma f(\mathcal{X}', \theta_0) + r)\right), \quad (13)$$

*where $\Theta_{xx'}$ is the neural tangent kernel of $f$.*

*Proof.* We begin by linearizing the function $f$ around its initialization parameters $\theta_0$:

$$f_{\text{lin}}(x, \theta_t) = f(x, \theta_0) + \nabla_\theta f(x, \theta_0)^\top (\theta_t - \theta_0). \quad (14)$$

We assume gradient descent updates with infinitesimal step size and a learning rate $\alpha$ on the loss

$$\mathcal{L}(\theta_t) = \tfrac{1}{2} \| \gamma f_{\text{lin}}(\mathcal{X}', \theta_t)_{\text{sg}} + r - f_{\text{lin}}(\mathcal{X}, \theta_t) \|_2^2, \quad (15)$$

yielding the parameter evolution

$$\frac{\mathrm{d}}{\mathrm{d}t}\theta_t = -\alpha \nabla_\theta \mathcal{L}(\theta_t). \quad (16)$$

Setting $w_t = \theta_t - \theta_0$ and find the learning dynamics:

$$\frac{\mathrm{d}}{\mathrm{d}t}w_t = -\alpha \nabla_\theta f(\mathcal{X}, \theta_0)\left(f_{\text{lin}}(\mathcal{X}, \theta_t) - (\gamma f_{\text{lin}}(\mathcal{X}', \theta_t) + r)\right). \quad (17)$$

Thus, the evolution of the linearized function is given by

$$\frac{\mathrm{d}}{\mathrm{d}t} f_{\text{lin}}(x, \theta_t) = -\alpha \nabla_\theta f(x, \theta_0)^\top \nabla_\theta f(\mathcal{X}, \theta_0)\left(f_{\text{lin}}(\mathcal{X}, \theta_t) - (\gamma f_{\text{lin}}(\mathcal{X}', \theta_t) + r)\right). \quad (18)$$

Letting $\delta_{\text{TD}}(\theta_t) = f_{\text{lin}}(\mathcal{X}, \theta_t) - (\gamma f_{\text{lin}}(\mathcal{X}', \theta_t) + r)$, we obtain the differential equation

$$\frac{\mathrm{d}}{\mathrm{d}t}\delta_{\text{TD}}(\theta_t) = -\alpha \left(\Theta_{\mathcal{X}\mathcal{X}}^{t_0} - \gamma\Theta_{\mathcal{X}'\mathcal{X}}^{t_0}\right)\delta_{\text{TD}}(\theta_t), \quad (19)$$

where $\Theta_{xx'}^{t_0} = \nabla_\theta f(x, \theta_0)^\top \nabla_\theta f(x', \theta_0)$ is the (empirical) tangent kernel of $f_{\text{lin}}(x, \theta_t)$. Since the linearization $f_{\text{lin}}(x, \theta_t)$ has constant gradients $\nabla_\theta f(x, \theta_0)$, the above differential equation is linear and solvable so long as the matrix $\Theta_{\mathcal{X}\mathcal{X}}^{t_0} - \gamma\Theta_{\mathcal{X}'\mathcal{X}}^{t_0}$ is positive definite. With an exponential ansatz, we obtain the solution

$$\delta_{\text{TD}}(\theta_t) = e^{-\alpha t \left(\Theta_{\mathcal{X}\mathcal{X}}^{t_0} - \gamma\Theta_{\mathcal{X}'\mathcal{X}}^{t_0}\right)}\delta_{\text{TD}}(\theta_0), \quad (20)$$

where $e^X$ is a matrix exponential. Reintegrating yields the explicit evolution of predictions

$$f_{\text{lin}}(x, \theta_t) = f(x, \theta_0) + \int_0^t \frac{\mathrm{d}}{\mathrm{d}t'} f_{\text{lin}}(x, \theta_{t'})\mathrm{d}t' \quad (21)$$

$$= f(x, \theta_0) - \Theta_{x\mathcal{X}}^{t_0}\left(\Theta_{\mathcal{X}\mathcal{X}}^{t_0} - \gamma\Theta_{\mathcal{X}'\mathcal{X}}^{t_0}\right)^{-1}\left(e^{-\alpha t\left(\Theta_{\mathcal{X}\mathcal{X}}^{t_0} - \gamma\Theta_{\mathcal{X}'\mathcal{X}}^{t_0}\right)} - I\right)\delta_{\text{TD}}(\theta_0). \quad (22)$$

Jacot et al. (2018) show that in the limit of infinite layer widths of the neural network, the NTK $\Theta_{xx'}^{t_0}$ becomes deterministic and constant $\Theta_{xx'}^{t_0} \to \Theta_{xx'}$. As a consequence, the linear approximation $f_{\text{lin}}(x; \theta_t)$ becomes exact w.r.t. the original function $\lim_{\text{width}\to\infty} f_{\text{lin}}(x; \theta_t) = f(x, \theta_t)$ (Lee et al., 2020).

$\square$

**Remark on the constancy of the NTK in TD learning.** We note here, that our proof assumed the results by Jacot et al. (2018) to hold for the case of semi-gradient TD updates, namely that the NTK becomes deterministic and constant $\Theta_{xx'}^{t_0} \to \Theta_{xx'}$ in the limit of infinite width under the here shown dynamics. First, the determinacy of the NTK at initialization follows from the law of large numbers and applies in our case equally as in the least squares case. The constancy of the NTK throughout training is established by Theorem 2 in Jacot et al. (2018), which we restate informally below.

**Theorem 2.** *[Jacot et al. (2018)] In the limit of infinite layer widths $n \to \infty$ and $n_1, \ldots, n_L = n$, the kernel $\Theta_{xx'}^{t_0}$ converges uniformly on the interval $t \in [0, T]$ to the constant neural tangent kernel*

$$\Theta_{xx'}^{t_0} \to \Theta_{xx'} ,$$

*provided that the integral $\int_0^T \|d_t\|_2 \, dt$ stays bounded. Here, $d_t \in \mathbb{R}^{N_D}$ is the training direction of the parameter evolution such that $\frac{d}{dt}\theta_t = -\alpha \nabla_\theta f(\mathcal{X}, \theta) d_t$*

In the here studied case of semi-gradient TD learning, the parameter evolution (as outlined above in Eq. (17)) is described by the gradient $\nabla_\theta f(\mathcal{X}, \theta_0)$ and the *training direction* $d_t$ according to

$$\frac{d}{dt}\theta_t = -\alpha \nabla_\theta f(\mathcal{X}, \theta_0) \underbrace{(f_{\text{lin}}(\mathcal{X}, \theta_t) - (\gamma f_{\text{lin}}(\mathcal{X}', \theta_t) + r))}_{d_t} , \tag{23}$$

where the training direction is given by $d_t = f_{\text{lin}}(\mathcal{X}, \theta_t) - (\gamma f_{\text{lin}}(\mathcal{X}', \theta_t) + r) = \delta_{TD}(\theta_t)$. Provided that the matrix $\Theta_{\mathcal{X}\mathcal{X}}^{t_0} - \gamma\Theta_{\mathcal{X}'\mathcal{X}}^{t_0}$ is positive definite, the norm of the training direction $\|d_t\|_2$ decays exponentially by Eq. 20. This implies

$$\|d_t\|_2 < \|d_0\|_2 e^{-t\lambda_{\min}} , \tag{24}$$

where $\lambda_{\min}$ is the smallest eigenvalue of $\Theta_{\mathcal{X}\mathcal{X}}^{t_0} - \gamma\Theta_{\mathcal{X}'\mathcal{X}}^{t_0}$. Assuming $\Theta_{\mathcal{X}\mathcal{X}}^{t_0} - \gamma\Theta_{\mathcal{X}'\mathcal{X}}^{t_0}$ is positive definite, $\lambda_{\min}$ is positive and as a consequence, we have

$$\int_0^\infty \|d_t\|_2 \, dt < \int_0^\infty \|d_0\|_2 e^{-t\lambda_{\min}} \, dt < \infty , \tag{25}$$

bounding the required integral of Theorem 2 for any $T$ and establishing $\Theta_{xx'}^{t_0} \to \Theta_{xx'}$ uniformly on the interval $[0, \infty)$ (see Theorem 2 in Jacot et al. (2018) for detailed proof for the last statement).

We note, however, that the condition for $\Theta_{\mathcal{X}\mathcal{X}}^{t_0} - \gamma\Theta_{\mathcal{X}'\mathcal{X}}^{t_0}$ to be positive definite is, for any $\gamma > 0$, stronger than in the classical results for supervised learning with least squares regression. While $\Theta_{\mathcal{X}\mathcal{X}}$ can be guaranteed to be positive definite for example by restricting $\mathcal{X}$ to lie on a unit-sphere, $x_i \in \mathcal{X}$ to be unique, and by assuming non-polynomial nonlinearities in the neural network (so as to prevent rank decay in the network expressivity), the condition is harder to satisfy in the TD learning setting. Here, the eigenspectrum of $\Theta_{\mathcal{X}\mathcal{X}}^{t_0} - \gamma\Theta_{\mathcal{X}'\mathcal{X}}^{t_0}$ tends to depend on the transitions $\mathcal{X} \to \mathcal{X}'$ themselves and thus is both dependent on the discount $\gamma$ as well as the interplay between gradient structures of the NTK and the MDP dynamics.

We note here, that this is not primarily a limitation of applying NTK theory to TD learning, but is reflected in practical experience: TD learning can, especially in offline settings, indeed be instable and diverge. Instability of this form is thus inherent to the learning algorithm rather than an artifact of our theoretical treatment. Informally, one approach towards guaranteeing positive definiteness of $\Theta_{\mathcal{X}\mathcal{X}}^{t_0} - \gamma\Theta_{\mathcal{X}'\mathcal{X}}^{t_0}$ is by enforcing diagonal dominance, appealing to the Gershgorin circle theorem (Gerschgorin, 1931). For a matrix $A = [a_{ij}]$, every real eigenvalue $\lambda$ must lie in

$$a_{ii} - R_i \leq \lambda \leq a_{ii} + R_i , \tag{26}$$

where $R_i = \sum_{i \neq j} |a_{ij}|$ is the sum of off-diagonal elements of a row $i$. In other words, a lower bound on the smallest real eigenvalue can be increased by increasing diagonal entries $a_{ii}$ while decreasing off-diagonal elements $a_{ij}$. In the TD learning setting, this translates to gradient conditioning, e.g., by ensuring $\|\nabla_\theta f(x, \theta)\|_2 = \|\nabla_\theta f(x', \theta)\|_2 = C$ for any pair $x, x'$, guaranteeing cross-similarities to be smaller than self-similarities. Indeed several recent works pursue similar strategies to stabilize offline TD learning (Yue et al., 2023; Gallici et al., 2024) and rely on architectural elements like layer normalization (Ba et al., 2016) to shape gradient norms.

A.1.2 POST TRAINING FUNCTION DISTRIBUTION WITH TEMPORAL DIFFERENCE DYNAMICS

We now aim to establish the distribution of post-training functions $f(x, t_\infty)$ when initial parameters $\theta_0$ are drawn randomly i.i.d. For the remainder of this section, we will assume the infinite width limit, s.t. $f_{\mathrm{lin}}(x, \theta_\infty) = f(x, \theta_\infty)$ and $\Theta_{xx'}^{t_0} = \Theta_{xx'}$. The post-training function $f(x, \theta_\infty)$ is given by

$$f(x, \theta_\infty) = f(x, \theta_0) - \Theta_{x\mathcal{X}} \left( \Theta_{\mathcal{X}\mathcal{X}}^{t_0} - \gamma \Theta_{\mathcal{X}'\mathcal{X}}^{t_0} \right)^{-1} \left( f(\mathcal{X}, \theta_0) - (\gamma f(\mathcal{X}', \theta_0) + r) \right), \quad (27)$$

and is thus a deterministic function of the initialization $\theta_0$.

**Theorem 1.** *Let $f(x, \theta_t)$ be a NN with $L$ hidden layers of width $n_1, \ldots, n_L = n$ trained with gradient flow to reduce the TD loss $\mathcal{L}(\theta_t) = \frac{1}{2} \| \gamma [f(\mathcal{X}', \theta_t)]_{sg} + r - f(\mathcal{X}, \theta_t) \|_2^2$. In the limit of infinite width $n \to \infty$ and time $t \to \infty$, the distribution of predictions $f(\mathcal{X}_T, \theta_\infty)$ on a set of test points $\mathcal{X}_T$ converges to a Gaussian with mean and covariance given by*

$$\mathbb{E}_{\theta_0} \left[ f(\mathcal{X}_T, \theta_\infty) \right] = \Theta_{\mathcal{X}_T \mathcal{X}} \Delta_{\mathcal{X}}^{-1} r,$$

$$\mathrm{Cov}_{\theta_0} \left[ f(\mathcal{X}_T, \theta_\infty) \right] = \kappa_{\mathcal{X}_T \mathcal{X}_T} - (\Theta_{\mathcal{X}_T \mathcal{X}} \Delta_{\mathcal{X}}^{-1} \Lambda_{\mathcal{X}_T} + h.c.) + (\Theta_{\mathcal{X}_T \mathcal{X}} \Delta_{\mathcal{X}}^{-1} (\Lambda_{\mathcal{X}} - \gamma \Lambda_{\mathcal{X}'}) \Delta_{\mathcal{X}}^{-1 \top} \Theta_{\mathcal{X}\mathcal{X}_T}),$$

*where $\Theta_{xx'}$ is the NTK, $\kappa_{xx'}$ is the NNGP kernel, h.c. denotes the Hermitian conjugate, and*

$$\Delta_{\tilde{\mathcal{X}}} = \Theta_{\mathcal{X}\tilde{\mathcal{X}}} - \gamma \Theta_{\mathcal{X}'\tilde{\mathcal{X}}}, \quad \text{and} \quad \Lambda_{\tilde{\mathcal{X}}} = \kappa_{\mathcal{X}\tilde{\mathcal{X}}} - \gamma \kappa_{\mathcal{X}'\tilde{\mathcal{X}}}.$$

*Proof.* We begin by introducing a column vector of post-training function evaluations on a set of test points $\mathcal{X}_T$, and the training data $\mathcal{X}$ and $\mathcal{X}'$. Moreover, we introduce the shorthand

$$\Delta_{\mathcal{X}} = \Theta_{\mathcal{X}\mathcal{X}} - \gamma \Theta_{\mathcal{X}'\mathcal{X}}, \quad (28)$$

and similarly $\Delta_{\mathcal{X}'} = \Theta_{\mathcal{X}\mathcal{X}'} - \gamma \Theta_{\mathcal{X}'\mathcal{X}'}$. The vector can then be compactly described in block matrix notation by

$$\underbrace{\begin{pmatrix} f(\mathcal{X}_T, \theta_\infty) \\ f(\mathcal{X}, \theta_\infty) \\ f(\mathcal{X}', \theta_\infty) \end{pmatrix}}_{f^\infty} = \underbrace{\begin{pmatrix} I & -\Theta_{\mathcal{X}_T \mathcal{X}} \Delta_{\mathcal{X}}^{-1} & \gamma \Theta_{\mathcal{X}_T \mathcal{X}} \Delta_{\mathcal{X}}^{-1} \\ I & -\Theta_{\mathcal{X}\mathcal{X}} \Delta_{\mathcal{X}}^{-1} & \gamma \Theta_{\mathcal{X}\mathcal{X}} \Delta_{\mathcal{X}}^{-1} \\ I & -\Theta_{\mathcal{X}'\mathcal{X}} \Delta_{\mathcal{X}}^{-1} & \gamma \Theta_{\mathcal{X}'\mathcal{X}} \Delta_{\mathcal{X}}^{-1} \end{pmatrix}}_{A} \underbrace{\begin{pmatrix} f(\mathcal{X}_T, \theta_0) \\ f(\mathcal{X}, \theta_0) \\ f(\mathcal{X}', \theta_0) \end{pmatrix}}_{f^0} + \underbrace{\begin{pmatrix} \Theta_{\mathcal{X}_T \mathcal{X}} \Delta_{\mathcal{X}}^{-1} r \\ \Theta_{\mathcal{X}\mathcal{X}} \Delta_{\mathcal{X}}^{-1} r \\ \Theta_{\mathcal{X}'\mathcal{X}} \Delta_{\mathcal{X}}^{-1} r \end{pmatrix}}_{b}. \quad (29)$$

Lee et al. (2018) show that neural networks with random Gaussian initialization $\theta_0$ (including NTK parametrization) are described by the neural network Gaussian process (NNGP) $f(\mathcal{X}_T, \theta_0) \sim \mathcal{N}(0, \kappa_{\mathcal{X}_T \mathcal{X}_T})$ with $\kappa_{\mathcal{X}_T \mathcal{X}_T} = \mathbb{E}[f(\mathcal{X}_T, \theta_0) f(\mathcal{X}_T, \theta_0)^\top]$. By extension, the initializations $f^0$ are jointly Gaussian with zero mean and covariance matrix

$$\mathrm{Cov}[f^0] = \underbrace{\begin{pmatrix} \kappa_{\mathcal{X}_T \mathcal{X}_T} & \kappa_{\mathcal{X}_T \mathcal{X}} & \kappa_{\mathcal{X}_T \mathcal{X}'} \\ \kappa_{\mathcal{X}\mathcal{X}_T} & \kappa_{\mathcal{X}\mathcal{X}} & \kappa_{\mathcal{X}\mathcal{X}'} \\ \kappa_{\mathcal{X}'\mathcal{X}_T} & \kappa_{\mathcal{X}'\mathcal{X}} & \kappa_{\mathcal{X}'\mathcal{X}'} \end{pmatrix}}_{K}. \quad (30)$$

As the post-training function evaluations $f^\infty$ given in Eq. (29) are affine transformations of the multivariate Gaussian random variables $f^0 \sim \mathcal{N}(0, K)$, they themselves are multivariate Gaussian with distribution $f^\infty \sim \mathcal{N}(b, A K A^\top)$.

We are content with obtaining an expression for the distribution of $f(\mathcal{X}_T, \theta_\infty)$ and thus in the following focus on the top-left entry of the block matrix $(A K A^\top)_{11}$. For notational brevity, we introduce the following shorthand notations

$$\Lambda_{\tilde{\mathcal{X}}} = \kappa_{\mathcal{X}\tilde{\mathcal{X}}} - \gamma \kappa_{\mathcal{X}'\tilde{\mathcal{X}}} \quad (31)$$

After some rearranging, one obtains the following expression for the covariance $\mathrm{Cov}(f_{\mathcal{X}_T}^\infty)$

$$\mathrm{Cov}_{\theta_0} \left[ f(\mathcal{X}_T, \theta_\infty) \right] = \kappa_{\mathcal{X}_T \mathcal{X}_T} - (\Theta_{\mathcal{X}_T \mathcal{X}} \Delta_{\mathcal{X}}^{-1} \Lambda_{\mathcal{X}_T} + h.c.) + (\Theta_{\mathcal{X}_T \mathcal{X}} \Delta_{\mathcal{X}}^{-1} (\Lambda_{\mathcal{X}} - \gamma \Lambda_{\mathcal{X}'}) \Delta_{\mathcal{X}}^{-1 \top} \Theta_{\mathcal{X}\mathcal{X}_T}).$$

$\square$

### A.1.3 DISTRIBUTION OF UVU PREDICTIVE ERRORS

We now aim to find an analytical description of the predictive errors as generated by our approach. For this, let $u(x, \vartheta_t)$ denote the predictive (online) network and $g(x; \psi_0)$ the fixed target network. We furthermore denote $\epsilon(x, \vartheta_t, \psi_0) = u(x, \vartheta_t) - g(x, \psi_0)$ the prediction error between online and target network.

**Corollary 1.** *Under the conditions of Theorem 1, let $u(x, \vartheta_\infty)$ be a converged online predictor trained with synthetic rewards generated by the fixed target network $g(x, \psi_0)$ with $r_g^z = g(\mathcal{X}, \psi_0) - \gamma g(\mathcal{X}', \psi_0)$. Furthermore denote the variance of converged universal Q-functions $\mathbb{V}_{\theta_0}[Q(x, \theta_\infty)]$. Assume $u, g,$ and $Q$ are architecturally equal and parameters are drawn i.i.d. $\theta_0, \vartheta_0, \psi_0 \sim \mathcal{N}(0, 1)$. The expected squared prediction error coincides with Q-function variance*

$$\mathbb{E}_{\vartheta_0, \psi_0}\left[\tfrac{1}{2}\epsilon(x, \vartheta_\infty, \psi_0)^2\right] = \mathbb{V}_{\theta_0}[Q(x, \theta_\infty)], \tag{11}$$

*where the l.h.s. expectation and r.h.s. variance are taken over random initializations $\vartheta_0, \psi_0, \theta_0$.*

*Proof.* Since our algorithm uses semi-gradient TD losses to train $u(x, \vartheta_t)$, the linearized dynamics of Theorem (1) apply. However, we consider a fixed target network $g(x; \psi_0)$ to produce synthetic rewards according to

$$r_g = g(x, \psi_0) - \gamma g(x', \psi_0). \tag{32}$$

With the post training function as described by Eq. 27, the post-training prediction error in a query point $x$ for this reward is given by

$$u(x, \vartheta_\infty) - g(x, \psi_0) =$$
$$u(x, \vartheta_0) - g(x, \psi_0) - \Theta_{x\mathcal{X}}\Delta_{\mathcal{X}}^{-1}(u(\mathcal{X}, \vartheta_0) - (\gamma u(\mathcal{X}', \vartheta_0) + g(\mathcal{X}, \psi_0) - \gamma g(\mathcal{X}', \psi_0))). \tag{33}$$

We again use the shorthand $\epsilon^t = (\epsilon(\mathcal{X}_T, \vartheta_t, \psi_0), \epsilon(\mathcal{X}, \vartheta_t, \psi_0), \epsilon(\mathcal{X}', \vartheta_t, \psi_0))^\top$ and reusing the block matrix $A$ from Eq. 29, we can write

$$\epsilon^\infty = A\epsilon^0. \tag{34}$$

By assumption, $u(x, \vartheta_0)$ and $g(x, \psi_0)$ are architecturally equivalent and initialized i.i.d., and $\epsilon^0$ is simply the sum of two independent Gaussian vectors with covariance $\mathrm{Cov}[\epsilon^0] = 2K$. We conclude that prediction errors $\epsilon^\infty$ are Gaussian with distribution $\epsilon^\infty \sim \mathcal{N}(0, 2AKA^\top)$. Taking the diagonal of the covariance matrix $AKA_{11}^\top$, we obtain

$$\mathbb{E}_{\vartheta_0, \psi_0}\left[\tfrac{1}{2}\epsilon(x, \vartheta_\infty, \psi_0)^2\right] = \mathbb{V}_{\theta_0}[Q(x, \theta_\infty)], \tag{35}$$

where

$$\mathbb{V}_{\theta_0}[Q(x, \theta_\infty)] = \kappa_{xx} - (\Theta_{x\mathcal{X}}\Delta_{\mathcal{X}}^{-1}\Lambda_x + h.c.) + (\Theta_{x\mathcal{X}}\Delta_{\mathcal{X}}^{-1}(\Lambda_{\mathcal{X}} - \gamma\Lambda_{\mathcal{X}'})\Delta_{\mathcal{X}}^{-1^\top}\Theta_{\mathcal{X}x}). \tag{36}$$

$\square$

## A.2 MULTIHEADED UVU

We now show results concerning the equivalence of multiheaded UVU prediction errors and finite ensembles of Q-functions. We first outline proofs for two results by Lee et al. (2018) and Jacot et al. (2018), which rely on in our analysis.

### A.2.1 NEURAL NETWORK GAUSSIAN PROCESS PROPAGATION AND INDEPENDENCE

Consider a deep neural network $f$ with $L$ layers. Let $z_i^l(x)$ denote the $i$-th output of layer $l = 1, \ldots, L$, defined recursively as:

$$z_i^l(x) = \sigma_b b_i^l + \frac{\sigma_w}{\sqrt{n_{l-1}}}\sum_{j=1}^{n_{l-1}} w_{ij}^l x_j^l(x), \quad x_j^l(x) = \phi(z_j^{l-1}(x)), \tag{37}$$

where $n_l$ is the width of layer $l$ with $n_0 = n_{\text{in}}$ and $x^0 = x$. Further, $\sigma_w$ and $\sigma_b$ are constant variance multipliers, weights $w^l$ and biases $b^l$ are initialized i.i.d. with $\mathcal{N}(0, 1)$, and $\phi$ is a Lipschitz-continuous nonlinearity. The $i$-th function output $f_i(x)$ of the NN is then given by $f_i(x) = z_i^L(x)$.

**Proposition 2** (Lee et al. (2018)). *At initialization and in the limit $n_1 \ldots, n_{L-1} \to \infty$, the $i$-th output at layer $l$, $z_i^l(x)$, converges to a Gaussian process with zero mean and covariance function $\kappa_{ii}^l$ given by*

$$\kappa_{ii}^1(x, x') = \frac{\sigma_w^2}{n_0} x^\top x' + \sigma_b^2, \quad and \quad k_{ij}^1 = 0, \quad i \neq j. \tag{38}$$

$$\kappa_{ii}^l(x, x') = \sigma_b^2 + \sigma_w^2 \mathbb{E}_{z_i^{l-1} \sim \mathcal{GP}(0, \kappa_{ii}^{l-1})}[\phi(z_i^{l-1}(x))\phi(z_i^{l-1}(x'))]. \tag{39}$$

$$\tag{40}$$

*and*

$$\kappa_{ij}^l(x, x') = \mathbb{E}[z_i^l(x)z_j^l(x')] = \begin{cases} \kappa^l(x, x') & \text{if } i = j, \\ 0 & \text{if } i \neq j. \end{cases} \tag{41}$$

*Proof.* The proof is done by induction. The induction assumption is that if outputs at layer $l - 1$ satisfy a GP structure

$$z_i^{l-1} \sim \mathcal{GP}(0, \kappa_{ii}^{l-1}), \tag{42}$$

with the covariance function defined as

$$\kappa_{ii}^{l-1}(x, x') = \mathbb{E}[z_i^{l-1}(x)z_i^{l-1}(x')] = k_{jj}^{l-1}(x, x'), \quad \forall i, j, \tag{43}$$

$$\kappa_{ij}^{l-1}(x, x') = \mathbb{E}[z_i^{l-1}(x)z_j^{l-1}(x')] = 0, \quad \text{for } i \neq j, \tag{44}$$

then, outputs at layer $l$ follow

$$z_i^l(x) \sim \mathcal{GP}(0, \kappa_{ii}^l), \tag{45}$$

where the kernel at layer $l$ is given by:

$$\kappa_{ii}^l(x, x') = \mathbb{E}[z_i^l(x)z_i^l(x')] = \kappa_{jj}^l(x, x'), \quad \forall i, j, \tag{46}$$

$$\kappa_{ij}^l(x, x') = \mathbb{E}[z_i^l(x)z_j^l(x')] = 0, \quad \text{if } i \neq j. \tag{47}$$

with the recursive definition

$$\kappa_{ii}^l(x, x') = \sigma_b^2 + \sigma_w^2 \mathbb{E}_{z_i^{l-1} \sim \mathcal{GP}(0, k_{ii}^{l-1})}[\phi(z_i^{l-1}(x))\phi(z_i^{l-1}(x'))]. \tag{48}$$

*Base case* ($l = 1$). At layer $l = 1$ we have:

$$z_i^1(x) = \frac{\sigma_w}{\sqrt{n_0}} \sum_{j=1}^{n_0} w_{ij}^1 x_j + \sigma_b b_i^1. \tag{49}$$

This is an affine transform of Gaussian random variables; thus, $z_i^1(x)$ is Gaussian distributed with

$$z_i^1(x) \sim \mathcal{GP}(0, \kappa_{ii}^1), \tag{50}$$

with kernel

$$\kappa_{ii}^1(x, x') = \frac{\sigma_w^2}{n_0} x^\top x' + \sigma_b^2, \quad \text{and} \quad \kappa_{ij}^1 = 0, \quad i \neq j. \tag{51}$$

*Induction step* $l > 1$. For layers $l > 1$ we have

$$z_i^l(x) = \sigma_b b_i^l + \frac{\sigma_w}{\sqrt{n_{l-1}}} \sum_{j=1}^{n_{l-1}} w_{ij}^l x_j^l(x), \quad x_j^l(x) = \phi(z_j^{l-1}(x)). \tag{52}$$

By the induction assumption, $z_j^{l-1}(x)$ are generated by independent Gaussian processes. Hence, $x_i^l(x)$ and $x_j^l(x)$ are independent for $i \neq j$. Consequently, $z_i^l(x)$ is a sum of independent random variables. By the Central Limit Theorem (as $n_1, \ldots, n_{L-1} \to \infty$) the tuple $\{z_i^l(x), z_i^l(x')\}$ tends to be jointly Gaussian, with covariance given by:

$$\mathbb{E}[z_i^l(x)z_i^l(x')] = \sigma_b^2 + \sigma_w^2 \mathbb{E}_{z_i^{l-1} \sim \mathcal{GP}(0, \kappa_{ii}^{l-1})}[\phi(z_i^{l-1}(x))\phi(z_i^{l-1}(x'))]. \tag{53}$$

Moreover, as $z_i^l$ and $z_j^l$ for $i \neq j$ are defined through independent rows of the parameters $w^l, b^l$ and independent pre-activations $x^l(x)$, we have

$$\kappa_{ij}^l = \mathbb{E}[z_i^l(x)z_j^l(x')] = 0, \quad i \neq j, \tag{54}$$

completing the proof. $\square$

A.2.2 NEURAL TANGENT KERNEL PROPAGATION AND INDEPENDENCE

We change notation slightly from the previous section to make the parametrization of $f_i(x, \theta^L)$ and $z_i^l(x; \theta^l)$ explicit with

$$z_i^l(x, \theta^l) = \sigma_b b_i^l + \frac{\sigma_w}{\sqrt{n_{l-1}}} \sum_{j=1}^{n_{l-1}} w_{ij}^l x_j^l(x), \quad x_j^l(x) = \phi(z_j^{l-1}(x; \theta^{l-1})), \tag{55}$$

where $\theta^l$ denotes the parameters $\{w^1, b^1, \dots, w^l, b^l\}$ up to layer $l$ and $f_i(x, \theta^L) = z_i^L(x; \theta^L)$. Let furthermore $\phi$ be a Lipschitz-continuous nonlinearity with derivative $\dot{\phi}(x) = \frac{d}{dx}\phi(x)$.

**Proposition 3** (Jacot et al. (2018)). *In the limit $n_1 \dots, n_{L-1} \to \infty$, the neural tangent kernel $\Theta_{ii}^l(x, x')$ of the $i$-th output $z_i^l(x, \theta^l)$ at layer $l$, defined as the gradient inner product*

$$\Theta_{ii}^l(x, x') = \nabla_{\theta^l} z_i^l(x, \theta^l)^\top \nabla_{\theta^l} z_i^l(x', \theta^l), \tag{56}$$

*is given recursively by*

$$\Theta_{ii}^1(x, x') = \kappa_{ii}^1(x, x') = \frac{\sigma_w^2}{n_0} x^\top x' + \sigma_b^2, \quad \text{and} \quad \Theta_{ij}^1(x, x') = 0, \quad i \neq j. \tag{57}$$

$$\Theta_{ii}^l(x, x') = \Theta_{ii}^{l-1}(x, x')\dot{\kappa}_{ii}^{l-1}(x, x') + \kappa_{ii}^l(x, x'), \tag{58}$$

$$\tag{59}$$

*where*

$$\dot{\kappa}_{ii}^l(x, x') = \sigma_w^2 \mathbb{E}_{z_i^{l-1} \sim \mathcal{GP}(0, \kappa_{ii}^{l-1})}[\dot{\phi}(z_i^{l-1}(x))\dot{\phi}(z_i^{l-1}(x'))] \tag{60}$$

*and*

$$\Theta_{ij}^l(x, x') = \nabla_{\theta^l} z_i^l(x, \theta^l)^\top \nabla_{\theta^l} z_j^l(x', \theta^l) = \begin{cases} \Theta^l(x, x') & \text{if } i = j, \\ 0 & \text{if } i \neq j. \end{cases} \tag{61}$$

*Proof.* We again proceed by induction. The induction assumption is that if gradients satisfy at layer $l-1$

$$\Theta_{ij}^{l-1}(x, x') = \nabla_{\theta^{l-1}} z_i^{l-1}(x, \theta^{l-1})^\top \nabla_{\theta^{l-1}} z_j^{l-1}(x', \theta^{l-1}) = \begin{cases} \Theta^{l-1}(x, x') & \text{if } i = j, \\ 0 & \text{if } i \neq j, \end{cases} \tag{62}$$

then at layer $l$ we have

$$\Theta_{ii}^l(x, x') = \Theta_{ii}^{l-1}(x, x')\dot{\kappa}_{ii}^l(x, x') + \kappa_{ii}^l(x, x') \tag{63}$$

and

$$\Theta_{ij}^l(x, x') = \nabla_{\theta^l} z_i^l(x, \theta^l)^\top \nabla_{\theta^l} z_j^l(x', \theta^l) = 0 \quad \text{if } i \neq j. \tag{64}$$

*Base Case ($l = 1$).* At layer $l = 1$, we have

$$z_i^1(x) = \sigma_b b_i^1 + \frac{\sigma_w}{\sqrt{n_0}} \sum_j^{n_0} w_{ij}^1 x_j, \tag{65}$$

and the gradient inner product is given by:

$$\nabla_{\theta^1} z_i^1(x, \theta^1)^\top \nabla_{\theta^1} z_i^1(x', \theta^1) = \frac{\sigma_w^2}{n_0} x^\top x' + \sigma_b^2 = \kappa_{ii}^1(x, x'). \tag{66}$$

*Inductive Step ($l > 1$).* For layers $l > 1$, we split parameters $\theta^l = \theta^{l-1} \cup \{w^l, b^l\}$ and split the inner product by

$$\Theta_{ii}^l(x, x') = \underbrace{\nabla_{\theta^{l-1}} z_i^l(x, \theta^l)^\top \nabla_{\theta^{l-1}} z_i^l(x', \theta^l)}_{l.h.s} + \underbrace{\nabla_{\{w^l, b^l\}} z_i^l(x, \theta^l)^\top \nabla_{\{w^l, b^l\}} z_i^l(x', \theta^l)}_{r.h.s}. \tag{67}$$

Note that the $r.h.s$ involves gradients w.r.t. last-layer parameters, i.e. the post-activation outputs of the previous layer, and by the same arguments as in the NNGP derivation of Proposition 2, this is a sum of independent post activations s.t. in the limit $n_{l-1} \to \infty$

$$\nabla_{\{w^l, b^l\}} z_i^l(x, \theta^l)^\top \nabla_{\{w^l, b^l\}} z_j^l(x', \theta^l) = \begin{cases} k_{ii}^l(x, x'), & i = j, \\ 0, & i \neq j. \end{cases} \tag{68}$$

For the $l.h.s.$, we first apply chain rule to obtain

$$\nabla_{\theta^{l-1}} z_i^l(x, \theta^l) = \frac{\sigma_w}{\sqrt{n_{l-1}}} \sum_j^{n_{l-1}} w_{ij}^l \dot{\phi}(z_j^{l-1}(x, \theta^{l-1})) \nabla_{\theta^{l-1}} z_j^{l-1}(x, \theta^{l-1}). \tag{69}$$

The gradient inner product of outputs $i$ and $j$ thus reduces to

$$\nabla_{\theta^{l-1}} z_i^l(x, \theta^l)^\top \nabla_{\theta^{l-1}} z_j^l(x', \theta^l) =$$

$$\frac{\sigma_w^2}{n_{l-1}} \sum_k^{n_{l-1}} w_{ik}^l w_{jk}^l \dot{\phi}(z_k^{l-1}(x, \theta^{l-1})) \dot{\phi}(z_k^{l-1}(x', \theta^{l-1})) \Theta_{kk}^{l-1}(x, x'). \tag{70}$$

By the induction assumption $\Theta_{kk}^{l-1}(x, x') = \Theta^{l-1}(x, x')$ and again by the independence of the rows $w_i^l$ and $w_j^l$ for $i \neq j$, the above expression converges in the limit $n_{l-1} \to \infty$ to an expectation with

$$\Theta_{ij}^l(x, x') = \begin{cases} \Theta^{l-1}(x, x') \dot{\kappa}_{ii}^l(x, x') + \kappa_{ii}^l(x, x') & i = j, \\ 0 & i \neq j. \end{cases} \tag{71}$$

This completes the induction. $\qquad\square$

### A.2.3 MULTIHEADED UVU: FINITE SAMPLE ANALYSIS

We now define multiheaded predictor with $M$ output heads $u_i(x, \vartheta_t)$ for $i = 1, \ldots, M$ and a fixed multiheaded target network $g_i(x_t; \psi_0)$ of equivalent architecture as $u$ with the corresponding prediction error $\epsilon_i(x, \vartheta_t, \psi_0)$ accordingly. Let $u_i(x, \vartheta_t)$ be trained such that each head runs the same algorithm as outlined in Section 3 independently.

**Corollary 2.** *Under the conditions of Theorem 1, consider online and target networks with $M$ independent heads $u_i, g_i$, $i = 1, \ldots, M$, each trained to convergence with errors $\epsilon_i(x, \vartheta_\infty, \psi_0)$. Let $\frac{1}{2}\bar{\epsilon}(x, \vartheta_\infty, \psi_0)^2 = \frac{1}{2M} \sum_{i=1}^M \epsilon_i(x, \vartheta_\infty, \psi_0)^2$ be the sample mean squared prediction error over $M$ heads. Moreover, consider $M + 1$ independent converged $Q$-functions $Q_i(x; \theta_\infty)$ and denote their sample variance $\bar{\sigma}_Q^2(x, \theta_\infty) = \frac{1}{M} \sum_{i=1}^{M+1} (Q_i(x; \theta_\infty) - \bar{Q}(x; \theta_\infty))^2$, where $\bar{Q}$ is the sample mean. The two estimators are identically distributed according to a scaled Chi-squared distribution*

$$\frac{1}{2}\bar{\epsilon}(x, \vartheta_\infty, \psi_0)^2 \stackrel{D}{=} \bar{\sigma}_Q^2(x, \theta_\infty), \quad \bar{\sigma}_Q^2(x, \theta_\infty) \sim \frac{\sigma_Q^2}{M} \chi^2(M), \tag{12}$$

*with $M$ degrees of freedom and $\sigma_Q^2(x, \theta_\infty) = \mathbb{V}_{\theta_0}[Q(x, \theta_\infty)]$ is the analytical variance of converged $Q$-functions given by Theorem 1.*

*Proof.* By Collorary. 1, the prediction error of a single headed online and target network $\epsilon(x, \vartheta_t, \psi_0) = u(x, \vartheta_t) - g(x, \psi_0)$ converges in the limit $n_1 \ldots, n_{L-1} \to \infty$ and $t \to \infty$ to a Gaussian with zero mean and variance $\epsilon(x, \vartheta_\infty, \psi_0) \sim \mathcal{N}(0, 2\sigma_Q^2)$ where

$$\sigma_Q^2 = \mathbb{V}_{\theta_0}[Q(x, \theta_\infty)] = \kappa_{xx} - (\Theta_{x\mathcal{X}} \Delta_{\mathcal{X}}^{-1} \Lambda_x + h.c.) + (\Theta_{x\mathcal{X}} \Delta_{\mathcal{X}}^{-1} (\Lambda_\mathcal{X} - \gamma \Lambda_{\mathcal{X}'}) \Delta_{\mathcal{X}}^{-1\top} \Theta_{\mathcal{X}x}). \tag{72}$$

By Propositions 2 and 3, the NNGP and NTK associated with each online head $u_i(x, \vartheta_\infty)$ in the infinite width and time limit are given by

$$\kappa_{ij}(x, x') = \mathbb{E}[u_i(x, \vartheta_\infty) u_j(x', \vartheta_\infty)] = \begin{cases} \kappa(x, x') & \text{if } i = j, \\ 0 & \text{if } i \neq j, \end{cases} \tag{73}$$

$$\Theta_{ij}(x, x') = \nabla_\vartheta u_i^l(x, \vartheta_\infty)^\top \nabla_\vartheta u_j^l(x', \vartheta_\infty) = \begin{cases} \Theta(x, x') & \text{if } i = j, \\ 0 & \text{if } i \neq j. \end{cases} \tag{74}$$

Due to the independence of the NNGP and NTK for different heads $u_i$, prediction errors $\epsilon_i(x_t; \vartheta_\infty, \psi_0)$ are i.i.d. draws from a zero mean Gaussian with variance equal as given in Eq. 72. Note that this is despite the final feature layer being shared between the output functions. The empirical mean squared prediction errors are thus Chi-squared distributed with $M$ degrees of freedom

$$\frac{1}{M} \sum_{i=1}^M \frac{1}{2} \epsilon_i(x_t; \vartheta_\infty, \psi_0)^2 \sim \frac{\sigma_Q^2}{M} \chi^2(M) \tag{75}$$

Now, let $\{Q_i(x;\theta_t)\}_{i=1}^{M+1}$ be a deep ensemble of $M+1$ Q-functions from independent initializations. By Corollary 1, these Q-functions, too, are i.i.d. draws from a Gaussian, now with mean $\Theta_{x\mathcal{X}}\Delta_{\mathcal{X}}^{-1}r$ and variance as given in Eq. 72. The sample variance of this ensemble thus also follows a Chi-squared distribution with $M$ degrees of freedom

$$\frac{1}{M}\sum_{i=1}^{M+1}\frac{1}{2}\left(Q_i(x;\theta_\infty)-\bar{Q}(x;\theta_\infty)\right)^2 \sim \frac{\sigma_Q^2}{M}\chi^2(M), \tag{76}$$

where $\bar{Q}(x;\theta_\infty) = \frac{1}{M+1}\sum_i^{M+1}Q_i(x;\theta_\infty)$ is the sample mean of $M+1$ universal Q-functions, completing the proof. $\square$

### A.3 LIMITATIONS AND ASSUMPTIONS

In this section, we detail central theoretical underpinnings and idealizations upon which our theoretical analysis is built.

A central element of our theoretical analysis is the representation of neural network learning dynamics via the Neural Tangent Kernel (NTK), an object in the theoretical limit of infinite network width. The established NTK framework, where the kernel is deterministic despite random initialziation and and constant throughout training, typically applies to fully connected networks with NTK parameterization, optimized using a squared error loss (Jacot et al., 2018). Our framework instead accommodates a semi-gradient TD loss, and thereby introduces an additional prerequisite for ensuring the convergence of these dynamics: the positive definiteness of the matrix expression $\Theta_{\mathcal{X}\mathcal{X}} - \gamma\Theta_{\mathcal{X}'\mathcal{X}}$. This particular constraint is more a characteristic inherent to the TD learning paradigm itself than a direct consequence of the infinite-width abstraction. Indeed, the design of neural network architectures that inherently satisfy such stability conditions for TD learning continues to be an active area of contemporary research (Yue et al., 2023; Gallici et al., 2024). The modeling choice of semi-gradient TD losses moreover does not incorporate the use of target networks, where bootstrapped values do not only stop gradients but are generated by a separate network altogether that slowly moves towards the online learner. Our analysis moreover considers the setting of *offline policy evaluation*, that is, we do not assume that additional data is acquired during learning and that policies evaluated for value learning remain constant. The assumption of a fixed, static dataset diverges from the conditions of online reinforcement learning with control, where the distribution of training data $(\mathcal{X}, \mathcal{X}')$ typically evolves as the agent interacts with its environment, both due to its collection of novel transitions and due to adjustments to the policy, for example by use of a Bellman optimality operator. Lastly, our theoretical model assumes, primarily for simplicity, that learning occurs under gradient flow with infinitesimally small step sizes and with updates derived from full-batch gradients. Both finite-sized gradient step sizes and stochastic minibatching has been treated in the literature, albeit not in the TD learning setting (Jacot et al., 2018; Lee et al., 2020; Liu et al., 2020; Yang, 2019). We believe our analysis could be extended to these settings without major modifications.

## B EXPERIMENTAL DETAILS

We provide details on our experimental setup, implementations and additional results. This includes architectural design choices, algorithmic design choices, hyperparameter settings, hyperparameter search procedures, and environment details.

### B.1 IMPLEMENTATION DETAILS

All algorithms are self-implemented and tuned in JAX (Bradbury et al., 2018). A detailed exposition of our design choices and parameters follows below.

**Environment Setup.** We use a variation of the GoToDoor environment of the minigrid suite (Chevalier-Boisvert et al., 2023). As our focus is not on partially observable settings, we use fully observable 35-dimensional state descriptions with $\mathcal{S} = \mathbb{R}^{35}$. Observation vectors comprise the factors:

$$o = \left(o_{\text{agent-pos}}^\top, o_{\text{agent-dir}}^\top, o_{\text{door-config}}^\top, o_{\text{door-pos}}^\top\right)^\top, \tag{77}$$

where $o_{\text{agent-pos}} \in \mathbb{R}^2$ is the agent position in $x, y$-coordinates, $o_{\text{agent-dir}} \in \mathbb{R}$ is a scalar integer indicating the agent direction (takes on values between 1 and 4), $o_{\text{door-config}} \in \mathbb{R}^{24}$ is the door configuration, comprising 4 one-hot encoded vectors indicating each door's color, and $o_{\text{door-pos}} \in \mathbb{R}^8$ is a vector containing the $x, y$-positions of the four doors. The action space is discrete and four-dimensional with the following effects

$$
a_{\text{effect}} = \begin{cases} \text{turn left} & \text{if } a = 0, \\ \text{turn right} & \text{if } a = 1, \\ \text{go forward} & \text{if } a = 2, \\ \text{open door} & \text{if } a = 3. \end{cases} \tag{78}
$$

Tasks are one-hot encodings of the target door color, that is $z \in \mathbb{R}^6$ and in the online setting are generated such that they are achievable. The reward function is an indicator function of the correct door being opened, in which case a reward of 1 is given to the agent and the agent position is reset to a random location in the grid. Episodes terminate only upon reaching the maximum number of timesteps (50 in our experiments).

In the task rejection setting described in our evaluation protocol, an agent in a start state $s_0$ is presented a list of tasks, which may or may not be attainable, and is allowed to reject a fixed number of tasks from this list. In our experiments, the agent is allowed to reject 4 out of 6 total tasks at the beginning of each episode.

**Data Collection.** Our offline datasets are recorded replay buffers from a DQN-agent deployed to the GoToDoor environment with an $\epsilon$-greedy exploration strategy and a particular policy: When the door indicated by the task encoding $z$ provided by the environment lies at the south or west wall, the regular policy by the online DQN agent is executed. If the target door lies at the north or east wall, however, actions are generated by a fixed random $Q$-network. This mixture policy emulates a policy that exhibits expert performance on certain combinations of tasks and states, but suboptimal behavior for other combinations. The replay buffer does, however, contain most combinations of states and tasks, albeit some with trajectories from suboptimal policies. Hyperparameter details of the online agent are provided in section B.2.

**Algorithmic Details.** All tested algorithms and experiments are based on DQN agents(Mnih et al., 2015) which we adapted for the task-conditioned universal value function (Schaul et al., 2015) setting. While our theoretical analysis considers full-batch gradient descent, in practice we sample minibatches from offline datasets with $\mathcal{X}_{mb} = \{(s_i, a_i, z_i)\}_{i=1}^{N_{mb}}$, $\mathcal{X}_b' = \{(s_i', a_i', z_i)\}_{i=1}^{N_b}$, where next-state actions are generated $a_i' = \arg\max_{a \in \mathcal{A}} Q(s_i', a, z_i, \theta_t)$ and rewards are $r = \{r_i\}_{i=1}^{N_{mb}}$. Moreover, we deviate from our theoretical analysis and use target networks in place of the stop-gradient operation. Here, a separate set of parameters $\tilde{\theta}_t$ is used to generate bootstrap targets in the TD loss which is in practice given by

$$
\mathcal{L}(\theta_t) = \tfrac{1}{2} \| \gamma Q(\mathcal{X}_{mb}', \tilde{\theta}_t) + r - Q(\mathcal{X}_{mb}, \theta_t) \|_2^2. \tag{79}
$$

The parameters $\tilde{\theta}_t$ are updated towards the online parameters $\theta_t$ at fixed intervals through polyak updating, as is common. We use this basic algorithmic pipeline for all tested algorithms, including the online agent used for data collection.

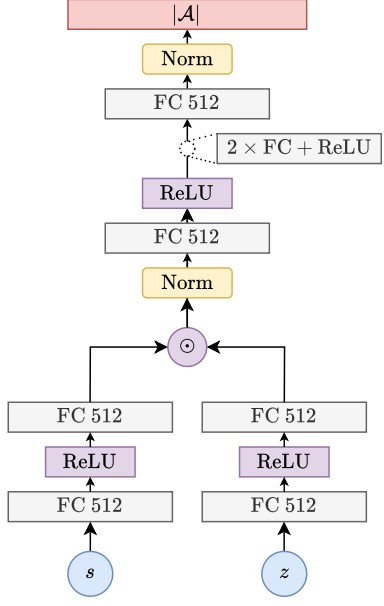

Figure 5: Illustration of the used architecture. $\odot$ indicates elementwise multiplication.

**Architectural Details.** We use a hypernet-
work MLP architecture adapted to the DQN setting, as depicted in Fig. 5. Specifically, this means we pass states $s$ and task encodings $z$ through single-layer encoders, which are then joint by elementwise multiplication. The resulting vector is normalized by its $l^2$ norm, $x' = \frac{x}{\|x\|_2}$. This joint vector is passed thorugh a 3-layer MLP with network width 512, again normalized by its $l^2$ norm and finally passed through a fully-connected layer to obtain a vector of dimension $\mathbb{R}^{|\mathcal{A}|}$. Although our experiments are conducted in the offline RL setting, preliminary experiments showed no benefits of using ensemble-based pessimism (An et al., 2021) or conservative $Q$-updates (Kumar et al., 2020). Instead, our normalization pipeline appears to sufficiently address overestimation issues as is suggested by several recent works (Yue et al., 2023; Gallici et al., 2024).

**Independent Bootstrapping.** For the ensemble-based BDQNP baseline and our UVU model, we perform independent bootstrapping in the TD loss computation. By this, we mean that both the bootstrapped value and actions are generated by individual $Q$-functions. In the case of BDQNP, this means we compute Loss 79 for each model $Q_k$, indexed by $k \in [1, \ldots, K]$ with $\mathcal{X}_{mb,k} = \mathcal{X}_{mb}$ and bootstraps are generated as

$$\mathcal{X}'_{mb,k} = \{(s'_i, a'_{ik}, z_i)\}_{i=1}^{N_{mb}}, \quad \text{and} \quad a'_{ik} = \text{argmax}_{a \in \mathcal{A}} Q_k(s'_i, a, z_i, \theta_t). \tag{80}$$

Note, that this procedure is established (Osband et al., 2016) and serves the purpose of maintaining independence between the models in the ensemble. In order to conduct the same procedure in our UVU method, where we have access to only one $Q$-function, we generate $K$ distinct $Q$-estimates by computing

$$Q_k^{UVU}(s, a, z, \theta_t) := Q(s, a, z, \theta_t) + \epsilon_k(s, a, z, \vartheta_t, \psi_0), \tag{81}$$

that is, by adding the UVU error of the $k$-th output head. Bootstraps are then generated according to Eq. 80.

**Intrinsic reward priors.** Intrinsic reward priors are a trick suggested by Zanger et al. (2024) to address a shortcoming of propagation methods used for intrinsic reward methods like RND(Burda et al., 2019; O'Donoghue et al., 2018). The issue is that while learning a $Q$-function with intrinsic rewards can, with the right choice of intrinsic reward, provide optimistic estimates of the value function, but only for state-action regions covered in the data. A potential underestiation of the optimistic bound, however, counteracts its intention, a phenomenon also described by Rashid et al. (2020). Intrinsic reward priors are a heuristic method to address this issue by adding local, myopic uncertainty estimates automatically to the forward pass of the intrinsic $Q$-function, leading to a "prior" mechanism that ensures a

$$\hat{Q}_{intr}(s, a, z, \theta_t) = Q_{intr}(s, a, z, \theta_t) + \tfrac{1}{2}\epsilon_{rnd}(s, a, z, \theta_{rnd})^2$$

where $\epsilon_{rnd}(s, a, z, \theta_{rnd})$ denotes a local RND error as an example. The altered function $\hat{Q}_{intr}(s, a, z, \theta_t)$ is trained as usual with Loss 79 and intrinsic rewards $\frac{1}{2}\epsilon_{rnd}(s, a, z, \theta_{rnd})^2$.

## B.2 HYPERPARAMETER SETTINGS

To ensure a consistent basis for comparison across our findings, all experimental work was carried out using a shared codebase. We adopted standardized modeling approaches, including uniform choices for elements like network architectures and optimization algorithms, where appropriate. Specifically, every experiment employed the same architecture as detailed in Appendix B.1. Key hyperparameters, encompassing both foundational and algorithm-specific settings, were tuned through a grid search on the $10 \times 10$ variation of the *GoToDoor* environment. The search grid and final hyperparamters are provided in Tables 2 and 3 respectively. DQN in Table 3 refers to the online data collection agent.

## B.3 ADDITIONAL EXPERIMENTAL RESULTS

We report additional results of the illustrative experiment shown in Section 3. Fig. 6 shows different uncertainty estimates in the described chain environment. The first row depicts *myopic* uncertainty estimates or, equivalently, RND errors. The second and third row show propagated local uncertainties with and without the intrinsic reward prior mechanism respectively. This result shows clearly the

Table 2: Hyperparameter search space

| Hyperparameter | Values |
|---|---|
| $Q$ Learning rate (all) | $[1 \cdot 10^{-6}, 3 \cdot 10^{-6}, 1 \cdot 10^{-5},$ $3 \cdot 10^{-5}, 1 \cdot 10^{-4}, 3 \cdot 10^{-4}, 1 \cdot 10^{-3}]$ |
| Prior function scale (BDQNP) | $[0.1, 0.3, 1.0, 3.0, 10.0]$ |
| RND Learning rate (RND, RND-P) | $[1 \cdot 10^{-6}, 3 \cdot 10^{-6}, 1 \cdot 10^{-5},$ $3 \cdot 10^{-5}, 1 \cdot 10^{-4}, 3 \cdot 10^{-4}, 1 \cdot 10^{-3}]$ |
| UVU Learning rate (UVU) | $[1 \cdot 10^{-6}, 3 \cdot 10^{-6}, 1 \cdot 10^{-5},$ $3 \cdot 10^{-5}, 1 \cdot 10^{-4}, 3 \cdot 10^{-4}, 1 \cdot 10^{-3}]$ |

Table 3: Hyperparameter settings for *GoToDoor* experiments.

| Hyperparameter | DQN | BDQNP | DQN-RND | DQN-RND+P | UVU |
|---|---|---|---|---|---|
| Adam $Q$-Learning rate | $3 \cdot 10^{-4}$ | $3 \cdot 10^{-4}$ | $3 \cdot 10^{-4}$ | $3 \cdot 10^{-4}$ | $3 \cdot 10^{-4}$ |
| Prior function scale | n/a | 1.0 | n/a | n/a | n/a |
| N-Heads 1 | 1 | 1 | 1 / 512 | 1 / 512 | 1 / 512 |
| Ensemble size | n/a | 3 / 15 | n/a | n/a | n/a |
| MLP hidden layers | 3 | | | | |
| MLP layer width | 512 | | | | |
| Discount $\gamma$ | 0.9 | | | | |
| Batch size | 512 | | | | |
| Adam epsilon | 0.005/batch size | | | | |
| Initialization | He uniform (He et al., 2015) | | | | |
| Double DQN | Yes (Hasselt, 2010) | | | | |
| Update frequency | 1 | | | | |
| Target lambda | 1.0 | | | | |
| Target frequency | 256 | | | | |

Table 4: GoToDoor Environment Settings

| Parameter | Value |
|---|---|
| State space dim | 35 |
| Action space dim | 3 |
| Task space dim | 6 |
| N Task Rejections | 4 |
| Max. Episode Length | 50 |

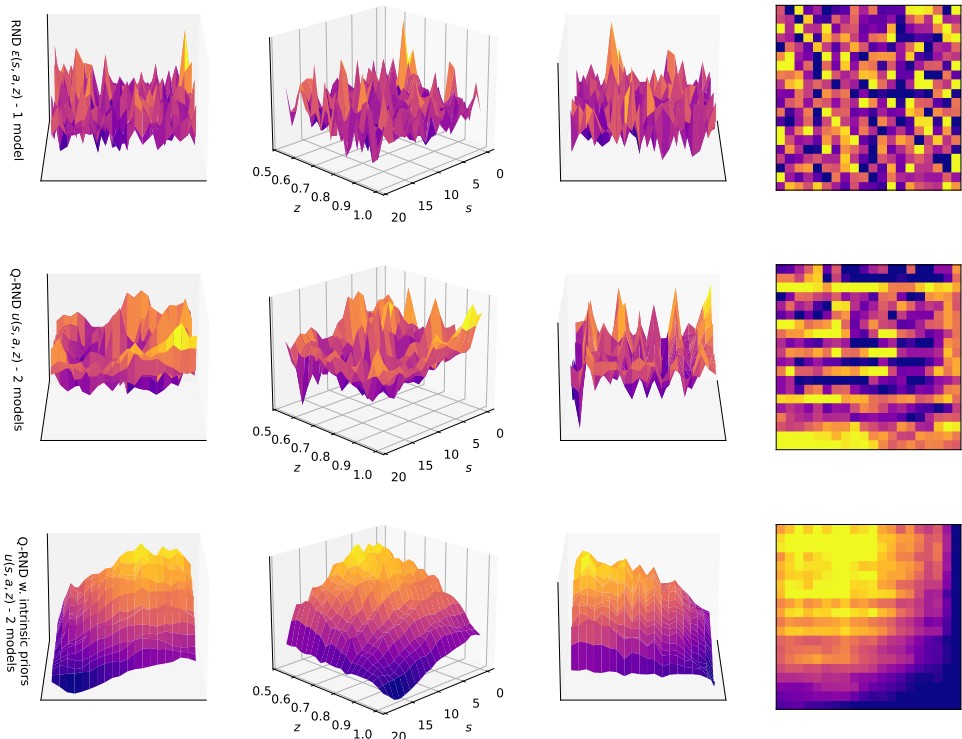

Figure 6: *Top Row*: RND errors. *2nd Row*: Value uncertainty as measured by an intrinsic $Q$-function. *3rd Row*: Value uncertainty as measured by an intrinsic $Q$-function with intrinsic reward priors.

shortcoming of the standard training pipeline for intrinsic rewards: in a standard training pipeline, the novelty bonus of RND is given only for transitions $(s_i, a_i, z_i, s_i')$ already present in the dataset and is never evaluated for OOD-actions. To generate reliable uncertainty estimates, RND requires, in addition to the RND network and the additional intrinsic $Q$-function, an algorithmic mechanism such as the *intrinsic reward priors* or even more sophisticated methods as described by Rashid et al. (2020).

## USE OF LARGE LANGUAGE MODELS

Large language models (LLMs) were used to assist in the preparation of this paper. Their usage was limited to refining sentence structure and verifying grammar, punctuation, and general language usage. No content or substantive research contributions were generated by LLMs.

