# OpenReview forum: "Universal Value-Function Uncertainties"
_ICLR.cc/2026/Conference — ICLR 2026 Poster_

### Official Review · Reviewer_1jtb · 2025-10-28

**Soundness:** 3
**Presentation:** 3
**Contribution:** 3
**Rating:** 6
**Confidence:** 3

**Summary:**

This paper proposes Universal Value-Function Uncertainties (UVU), a single network method for estimating policy-conditioned value uncertainty. The key idea is to train an online predictor via semi-gradient TD learning on synthetic rewards generated by a fixed random target network, and interpret the prediction error as an uncertainty measure. Using neural tangent kernel (NTK) theory, the authors derive post-training distributions of TD trained functions and show that, under the infinite-width and gradient-flow limit, the expected UVU prediction error is equivalent to the ensemble variance of universal $Q$ function. A multi-head version further yields a finite-sample equivalence to a scaled $\chi^2$ distribution.

**Strengths:**

- **Conceptual simplicity**: UVU replaces RND’s myopic novelty signal with policy-conditioned, future-aware value uncertainty. The concept of modeling TD error between learner and fixed random teacher provides an intuitive mechanism for epistemic uncertainty estimation.
- **Theoretical grounding**: The paper rigorously analyzes TD learning dynamics under NTK, presenting explicit mean-covariance forms for post-training predictions and proving the equivalence between UVU’s prediction error and ensemble variance.
- **Empirical effectiveness**: The proposed multi-head network achieves ensemble-level uncertainty quality while reducing computational cost by roughly an order of magnitude.

**Weaknesses:**

1. **Idealized stability condition**: Theoretical TD learning convergence condition (i.e., $\Theta_{XX}-\gamma\Theta_{X'X}\succ0$) depends on the transition structure and the choice of $\gamma$, which is difficult to verify or enforce in practice. Although the appendix suggests using Gershgorin’s theorem and gradient-norm conditioning to ensure diagonal dominance, it remains unclear whether this condition holds empirically or how violation manifests as instability (e.g., divergence of TD loss).
2. **Theory–implementation gap**: The theory assumes gradient flow, full-batch optimization, and stop-gradient TD, whereas the implementation employs mini-batches, target networks, and finite training steps. These deviations could introduce bias or affect convergence, yet no quantitative analysis is provided. In particular, the multi-head version uses a shared trunk, violating the independence assumption in Corollary 2 and potentially leading to systematic underestimation of uncertainty.
3. **Limited empirical generality**: Experiments are limited to a single offline MiniGrid setting. Evaluation on continuous control, stochastic, or partially observable environments would strengthen generalizability and robustness.

**Questions:**

1. Is there any observed correlation between the smallest eigenvalue of $\Theta_{XX}-\gamma\Theta_{X'X}$ and TD loss divergence?
2. How does the implementation gap such as inter-head correlations affect the uncertainty calibration?
3. Do you have plans to extend UVU to continuous control (e.g., D4RL) or stochastic gridworld environments?

---

> ### Author Response · Authors · 2025-11-19
> **Response to Reviewer 1jtb**
>
> **We thank the reviewer for their thoughtful and constructive review of our work and for highlighting both its conceptual contributions and theoretical strengths. We address the raised points below.**
>
> **W1: Stability in TD learning.**
>
> We agree that the convergence condition $\Theta_{\mathcal{X}\mathcal{X}} - \gamma \Theta_{\mathcal{X}'\mathcal{X}} \succ 0$ is a stricter condition in practice than in supervised learning. We note that, as discussed in Appendix A.1.1, this requirement is not specific to UVU but inherent to semi-gradient TD learning more broadly. This is reflected in practice, where RL algorithms are well-known to have a tendency to instability and divergence is a familiar phenomenon in TD-based methods. We also note that one could consider this intended behavior in UVU: if the TD value learner diverges, the associated predictive uncertainty should arguably likewise become unbounded, reflecting the fact that a diverged critic is arbitrarily uncertain about its predictions.
>
> **W2: Theory - implementation gap.**
>
> We appreciate the reviewer’s concern regarding deviations between our theoretical analysis in the NTK regime (gradient flow, full-batch optimization, semi-gradient TD) and the practical implementation (gradient descent, mini-batches, target networks, finite width). We focused empirically on width ablations (Fig. 4a), as network width is widely believed to be a dominant factor within the simplified assumptions of the NTK regime ([1-3]).
>
> Regarding the multiheaded architecture, we agree that shared trunks at finite width may introduce dependencies between heads. We note, however, that in the infinite width limit, the limiting GPs remain independent per-head even with shared architectures, so the assumption in Corollary 2 is not violated within the setting of our theoretical analysis. Empirically quantifying correlations in finite-width multiheaded architectures and their effect on uncertainty estimates is, indeed, an interesting research direction that could warrant a research undertaking in its own right. Our main experiment, however, provides evidence that these correlations do not prohibitively degrade uncertainty quality, albeit indirectly: multiheaded UVU performs on par with large independent ensembles on the task we study and thus did not suffer from a prohibitive degradation in predictive uncertainty quality.
>
> **W3: Empirical evaluation.**
>
> We acknowledge that applications of UVU in additional environments are of general interest. Within the scope of this submission, however, we prioritized the theoretical characterization of the algorithm and devised experiments that align with our theoretical setting. Specifically, our minigrid environments quantitatively assess the reliability of (multitask / universal) value uncertainty estimates while requiring algorithms and models to propagate long-term uncertainties and to maintain precise uncertainties for a multitude of policies. This accurately reflects our theoretical exposition and we thus consider further applications of UVU to a broader class of benchmarks and problems a promising direction for future work.
>
> **Q1: Correlation between eigenvalues and TD divergence.**
>
> Yes. In preliminary experiments, we observed a clear correlation between small (strongly negative) minimum eigenvalues of the TD transition Gram matrix $\Theta_{\mathcal{X}\mathcal{X}} - \gamma \Theta_{\mathcal{X}' \mathcal{X}}$ and diverging TD losses, although not all instances diverged to $\pm \infty$. This aligns with findings by Yue et al. [4], who specifically study divergence and the eigenspectrum of this matrix in particular.
>
> **Q2: Multiheaded architecture.**
>
> As noted above, correlations in the multiheaded architecture could indeed influence calibration of the uncertainty estimates. In practice, our main experiment suggests that such effects do not prevent UVU from matching the uncertainty quality of large independent ensembles: the model is able to estimate uncertainty reliably enough to solve a task that requires accurate uncertainty propagation. As stated earlier, we agree in principle that quantifying correlations in multiheaded architectures and their effect on uncertainty estimates is an interesting research direction in its own right but consider it beyond the scope of our current work.
>
> **Q3: Empirical evaluation.**
>
> UVU is not inherently restricted to discrete, deterministic or offline settings. While our current focus is on the theoretical characterization and the offline analysis (see W3), we consider applications of UVU to continuous or stochastic domains an interesting direction for future research.

---

> > ### Author Response · Authors · 2025-11-19
> > **References**
> >
> > [1] Lee, Jaehoon, et al. "Finite versus infinite neural networks: an empirical study." Advances in Neural Information Processing Systems 33 (2020): 1
> >
> > [2] Vyas, Nikhil, Yamini Bansal, and Preetum Nakkiran. "Limitations of the NTK for understanding generalization in deep learning, 2022." URL https://arxiv. org/abs/2206.10012.
> >
> > [3] Seleznova, Mariia, and Gitta Kutyniok. "Analyzing finite neural networks: Can we trust neural tangent kernel theory?." Mathematical and Scientific Machine Learning. PMLR, 2022.
> >
> > [4] Yue, Yang, et al. "Understanding, predicting and better resolving q-value divergence in offline-rl." Advances in Neural Information Processing Systems 36 (2023): 60247-60277.

---

### Official Review · Reviewer_4xcK · 2025-10-31

**Soundness:** 3
**Presentation:** 2
**Contribution:** 2
**Rating:** 4
**Confidence:** 3

**Summary:**

This paper introduces universal value-fucntion uncertainties (UVU), quantifying uncertainty as squared prediction errors between an online learner and a fixed, randomly initialized target network. The paper also provides a theoretical analysis of the proposed approach using neural tangent kernel (NTK) theory, showing that  UVU errors are equivalent to the variance of an ensemble of independent universal value functions in the limit of infinite network width.

**Strengths:**

The paper aims to address an important issue in RL, and the theoretical analysis is strong.

**Weaknesses:**

The central concern with the paper is the numerical study. Suppose that the true values are known for the simulated environment, the experiments should include coverage rates (and related calibration metrics). Reporting only the best policy’s returns with confidence intervals does not fully characterize estimator performance or uncertainty.

**Questions:**

My major concern is about the coverage rate of the true values. The authors may consider a simulated environment for which the true value functions are exactly known and demonstrate that the confidence intervals constructed by the proposed method can achieve the nomial coverage rate for the true value. Such an evaluation criterion is critically important for real decisions.

---

> ### Author Response · Authors · 2025-11-19
> **Response to Reviewer 4xcK**
>
> **We thank the reviewer for their time in reviewing our work and for their thoughtful and constructive feedback. We address each concern below.**
>
> **W1/Q1: Coverage rates.**
>
> We appreciate the reviewer’s suggestion to assess the method via empirical coverage rates against true value functions. Indeed, this is an interesting evaluation metric for uncertainty quantification in general. We note that such an evaluation, in the context of RL, requires exact knowledge of the ground-truth value function, which is typically difficult to obtain even in simulated environments. Exact coverage rates also lie somewhat beyond our theoretical claims, as these focus on the asymptotic equivalence between UVU and deep ensembles in the infinite-width NTK regime. This equivalence justifies UVU as an estimator of epistemic uncertainty as predicted by *ensemble variance*. However, this does not by itself imply or guarantee well-calibrated coverage without further assumptions. In practice, even ensemble methods typically require post-hoc calibration to achieve nominal coverage levels (see for example works [1-3] ). Still, we agree that assessing calibration - possibly via coverage of confidence intervals or expected calibration error and potentially including additional post-hoc calibration techniques - could provide a promising direction for future work.
>
> [1] Rahaman, Rahul. "Uncertainty quantification and deep ensembles." Advances in neural information processing systems 34 (2021): 20063-20075.
>
> [2] Zhang, Jize, Bhavya Kailkhura, and T. Yong-Jin Han. "Mix-n-match: Ensemble and compositional methods for uncertainty calibration in deep learning." International conference on machine learning. PMLR, 2020.
>
> [3] Kuleshov, Volodymyr, Nathan Fenner, and Stefano Ermon. "Accurate uncertainties for deep learning using calibrated regression." International conference on machine learning. PMLR, 2018.

---

### Official Review · Reviewer_DHG9 · 2025-10-31

**Soundness:** 3
**Presentation:** 3
**Contribution:** 2
**Rating:** 6
**Confidence:** 2

**Summary:**

This paper presents an approach for general uncertainty estimation of a value function, inspired by random network distillation. The authors present a theoretical motivation for their approach and showcase the effectiveness in an empirical analysis with a gridworld RL task.

This is a well-crafted paper, presenting a convincing and rigorous theoretical analysis. While strictly speaking, the novelty is limited, and the empirical analysis is restricted to a specialized setting, I still think it is a valuable contribution.

**Strengths:**

+ The paper is crafted very well, with high-quality writing, figures, and appendices
+ The topic of uncertainty estimation for RL (including for critics/value functions) remains relevant, with no widely accepted approaches. Existing techniques like deep ensembles or MC Dropout are available, but are not widespread (e.g., due to the additional computational demands or potential training instability), and MC Dropout in particular has been shown to provide mixed results
+ The motivation and preliminary sections are comprehensive and seem to cover all important aspects
+ The theoretical analysis utilizing NTK to show equality of uncertainty estimation to ensembles is convincing, and provides a strong justification of the approach

Neutral: The paper draws on existing work (RND and NTK). Compared to RND, it provides a theoretically grounded analysis and extension; compared to work on NTK, it is a novel application domain.

**Weaknesses:**

- The empirical analysis is somewhat limited (although it is an illustrative use case). I would love to see some additional experiments beyond this scenario. As the method is generalizable, I feel it should be possible to apply it to common benchmark environments (similar to the original RND paper)
- The lack of any implementation limits impact, and also makes it difficult to fully assess reproducibility.

**Questions:**

- You have only demonstrated the performance of UVU on an offline RL task. Do you expect UVU to behave differently in an online RL setting? During online training, the state distribution shifts, leading to a shifting critic as well (the critic network might be subject to catastrophic forgetting). Can this be accurately captured by UVU?
- During reading, i was strugling a bit with the introduction of the multi-head variant of UVU (maybe just due the clarity of writing). Could you strenghten the reasoning behind this analysis? In which case would a single UVU network no be sufficient as an estimator?

---

> ### Author Response · Authors · 2025-11-19
> **Response to Reviewer DHG9**
>
> **We thank the reviewer for their positive assessment and  thoughtful comments.**
>
> **W1 and W2: Empirical analysis and reproducibility.**
>
> For this submission, we chose to focus on a controlled setting that reflects core aspects of the algorithm and our theoretical analysis: the reliability of (multitask / universal) value uncertainty estimation which requires propagation, long-term credit / uncertainty assignment, and the ability to maintain precise uncertainties for a multitude of policies. We would also like to maintain that conceptual novelty and theoretical analysis are the main contribution of this work, rather than a wide outperforming of existing methods. Moreover, we will release a clean JAX implementation of UVU publically upon acceptance. We hope the reviewer understands that we typically do not release code for unpublished work, but we are committed to ensuring full reproducibility after acceptance.
>
> **Q1: Online RL.**
>
> UVU is not inherently restricted to offline settings. The algorithm itself can in principle be applied in online RL, where policies and state distribution evolve over time. However, our theoretical results are currently limited to the offline regime, where the data distribution is fixed, allowing us to derive closed-form learning dynamics in the NTK regime. Capturing the full dynamics of UVU in an online setting, i.e. under non-stationary data distributions and including phenomena like catastrophic forgetting, remains an open theoretical challenge - not just for our method, but for NTK-based analyses more broadly. That said, we do not expect practical barriers to applying UVU in online RL settings beyond those challenges faced by any standard deep RL pipeline.
>
> **Q2: Multi-headed UVU.**
>
> We thank the reviewer for highlighting a point of potential confusion. The multi-headed version aims to improve the reliability of the uncertainty estimates produced by UVU. Our first theoretical result regarding the predictive errors generated by UVU (Corollary 1) states that UVU errors match the analytical variance of a deep ensemble in expectation. Using multiple heads allows us to reduce the estimator variance of this expectation. Informally speaking, the multi-headed version then serves the purpose of producing error estimates that are more consistently close to this expectation than a single estimator would be. In the NTK-regime we in fact are able to show that an estimator using $M$ heads is statistically equivalent to a finite ensemble of $M+1$ deep neural networks (Corollary 2).

---

> ### Comment · Reviewer_DHG9 · 2025-11-24
>
> I thank the authors for their response. I think the reviewers agree in some common weaknesses, especially the lack of empirical results. However, i still feel that the paper, given the source code is published at acceptance, is a fair contribution. I will probably keep my score.

---

### Official Review · Reviewer_wiBU · 2025-11-01

**Soundness:** 2
**Presentation:** 3
**Contribution:** 3
**Rating:** 4
**Confidence:** 3

**Summary:**

The paper proposes a novel and computationally efficient single-network method (called "UVU") for estimating epistemic uncertainty in reinforcement learning. This approach measures uncertainty as the prediction error between an "online" network and a fixed, randomly initialized "target" network. The online network is trained using temporal difference learning on a synthetic reward signal generated by the target network itself, allowing the error to capture long-term value uncertainty. The paper provides a strong theoretical proof showing that, under infinite-width network assumptions, this single-model error is mathematically equivalent to the variance of a large, computationally expensive ensemble.

**Strengths:**

1. Improving the computational efficiency of epistemic uncertainty quantification is a key objective for RL algorithms.

2. The paper provides a formal proof, grounded in Neural Tangent Kernel (NTK) theory, that its single-network uncertainty estimate is mathematically equivalent to the variance of an infinite ensemble, moving it beyond a simple heuristic methods like Random Network Distillation (RND).

3. The policy-conditioned nature of the value function enables UVU to capture long-term, plan-dependent uncertainty. By making the policy representation an explicit input, the model is forced to learn a "universal" value function that understands how outcomes change under different plans. This design is crucial because it allows the temporal difference (TD) learning process to properly propagate uncertainty over time. If a specific policy leads to novel or data-sparse states far in the future, the TD updates naturally "back up" that high uncertainty to the agent's current state, providing a holistic, long-term risk assessment for that specific plan, rather than just a myopic "novelty" score of the current state.

**Weaknesses:**

**Lack of empirical results**

A potential weakness is the paper's limited evaluation of the intrinsic quality of the uncertainty estimates. The experiments focus on a binary task-rejection benchmark, which demonstrates the utility of the uncertainty but doesn't thoroughly analyze its calibration, its correlation with true value error across different states, or how it behaves in simpler, more controlled environments.

1. The paper compares its approach to Random Network Distillation (RND) and ensemble-based methods for epistemic uncertainty estimation, but it does not provide results on standard online RL benchmarks, where these methods have traditionally been evaluated. Including such results is crucial to demonstrate the practical utility of the learned uncertainty for efficient or safe exploration.

2. Because the method estimates policy-conditioned value uncertainty, its applicability in single-task settings—where the policy continuously evolves as the agent learns about the environment—remains unclear. This raises questions about the method’s feasibility in such scenarios.

**Questions:**

1. Could the authors provide results demonstrating that the method improves sample efficiency in online RL algorithms or facilitates safer exploration?

2. Is the method inherently limited to offline data and multi-task settings—where the number of policies is constrained and policy conditioning is easier to learn—compared to online RL, where the policy continually evolves as the agent learns the task?

---

> ### Author Response · Authors · 2025-11-19
> **Response to Reviewer wiBU**
>
> **We thank the reviewer for their thoughtful and constructive feedback. Below we address each point in turn.**
>
> **Q1: Empirical Results**
>
> We agree that additional applications of UVU - including in sample efficient online reinforcement learning - would be valuable in a general sense. The empirical analysis on minigrid provides a quantitative probe of the reliability of uncertainty estimates that require propagation, long-term credit (uncertainty) assignment, and the ability to maintain uncertainties for a multitude of policies and thus addresses our theoretical claims in a targeted way. We moreover emphasize that the primary contributions of the current work are conceptual novelty and theoretical analysis: UVU introduces a novel learning algorithm for estimating value uncertainty and is accompanied by a formal analysis that establishes its asymptotic equivalence to large ensembles under NTK dynamics. Due to the restricted timeframe in the current submission, online experiments are currently outside the scope of our work. But we believe the inherent algorithmic function of UVU is amenable to online RL as well and could motivate follow-up work that applies UVU to online exploration problems.
>
> **Q2: Applicability to online or single-task RL.**
>
> Algorithmically, UVU is not inherently tied to offline or multi-task settings. Our theoretical analysis, including our main equivalence results, are restricted to these regimes as they permit the closed-form solutions we derive to show UVU's asymptotic equivalence to deep ensembles. In practice, the algorithm can be deployed to online RL, for example by conditioning on an evolving policy representation as used in universal value functions, successor representations, or goal-conditioned RL. We consider its ability to condition on explicit policy representations a strength of UVU, as it enables principled uncertainty estimation in these settings (with existing methods often being less understood or computationally expensive). Still, UVU may in principle also be applied to single-task online RL learning (i.e., with an evolving policy and a trivial constant embedding $z$). However, we view these as implementation-level changes rather than fundamental limitations of the method.

---

### Comment · Area_Chair_Es4Y · 2025-11-22

Dear Reviewers,

The authors have responded to your reviews. Please review and respond to their comments.

Best,
Your AC

---

### Meta-Review · Area_Chair_2z71 · 2026-01-05

**Summary:**

Most reviewers are concerning about the limited empirical experiments and validations and some technical assumptions. Although the authors' rebuttal addresses and explains the assumption, it fails to provide additional experimental validations, which is not convincing from my aspect.

However, both AC and SAC agree and appreciate the theoretical contributions of this paper. Thus, I recommend acceptance and encourage authors to include additional experiment results for the camera ready version.

===

SAC note: The paper should be careful about its notation on value functions. The equation on L98 is problematic: what determines a Q-function is the full policy mapping, not the action distribution at a specific state as indicated by the superscript of $Q^{\pi(\cdot \mid s, z)}$. Also, $Q^\pi$ is typically reserved for the groundtruth value function and $Q$ without policy subscripts can refer to learned Q-functions, so $Q^\pi(\cdot, \theta)$ in Eq.(5) is very confusing.

Regarding reviewer's concern on coverage rates, I agree that the paper focuses on efficient approximation of ensemble variance which by itself does not offer coverage guarantee. That said, what you mentioned in the rebuttal, that "ground-truth value function... is typically difficult to obtain", is not quite accurate. For any given state-action pair, reset + Monte-Carlo rollouts can easily give you accurate Q-values.

**Reviewer Concerns:**

Reviewers wiBU and DHG9 concern about the lack of experiment validations and the applicability of the method in online learning. Reviewer 4xcK mainly concerns about the coverage rates. Reviewer 1jtb mainly questions on the technical assumptions and limited empirical results.

During the rebuttal, authors mostly address concerns on the technical assumptions. However, almost all reviewers are requesting additional experiment results for validation, yet authors defer it to future work and fail to provide enough validations. Thus I believe this major concern is not addressed.

**Reviewer Scores:**

Please see above part.

---

### Decision · Program_Chairs · 2026-01-26

Accept (Poster)